# Learning Lagrangian Interaction Dynamics with Sampling-Based Model Order Reduction

**Hrishikesh Viswanath**                                    *hviswan@purdue.edu*
*Purdue University*

**Yue Chang**                                    *changyue.chang@mail.utoronto.ca*
*University of Toronto*

**Aleksey Panas**                                    *aleksey.panas@alumni.utoronto.ca*
*University of Toronto*

**Julius Berner**                                    *jberner@nvidia.com*
*Nvidia*

**Peter Yichen Chen**                                    *pyc@csail.mit.edu*
*Massachusetts Institute of Technology*

**Aniket Bera**                                    *aniketbera@purdue.edu*
*Purdue University*

**Reviewed on OpenReview:** *https://openreview.net/forum?id=vXCQA1EzaG*

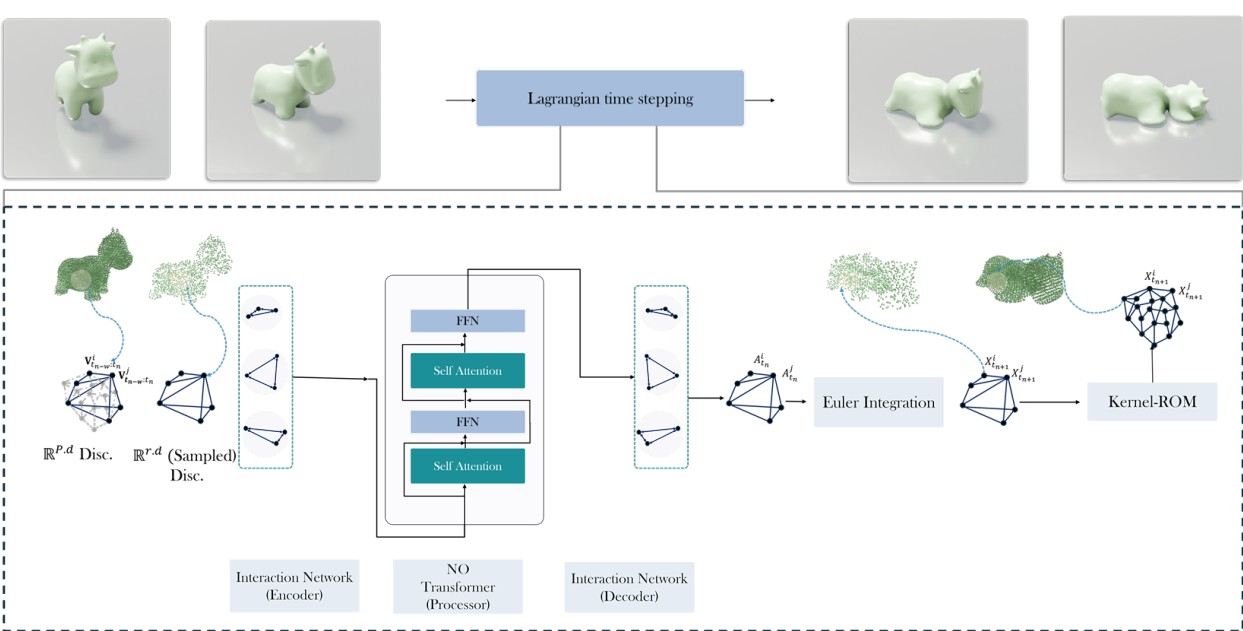

Figure 1: **Overview of GIOROM: Geometry-Informed Reduced-Order Modeling in physical space.** Rather than evolving a global latent representation, GIOROM advances a reduced set of Lagrangian particles directly in physical space using a neural autoregressive PDE operator $\phi_\Theta$. The operator predicts accelerations $\mathbf{A}_t$ from a short history of particle velocities $\mathbf{V}_{t-w:t}$, which are integrated to update particle positions. A learnable online Kernel based ROM parametrizes the solution manifold, enabling efficient querying at arbitrary spatial locations using local geometric information from the evolved particles.

## Abstract

Simulating physical systems governed by Lagrangian dynamics often entails solving partial differential equations (PDEs) over high-resolution spatial domains, leading to significant computational expense. Reduced-order modeling (ROM) mitigates this cost by evolving low-dimensional latent representations of the underlying system. While neural ROMs enable querying solutions from latent states at arbitrary spatial points, their latent states typically represent the global domain and struggle to capture localized, highly dynamic behaviors such as fluids. We propose a sampling-based reduction framework that evolves Lagrangian systems directly in physical space, over the particles themselves, reducing the number of active degrees of freedom via data-driven neural PDE operators. To enable querying at arbitrary spatial locations, we introduce a learnable kernel parameterization that uses local spatial information from time-evolved sample particles to infer the underlying solution manifold. Empirically, our approach achieves a $6.6\times$–$32\times$ reduction in input dimensionality while maintaining high-fidelity evaluations across diverse Lagrangian regimes, including fluid flows, granular media, and elastoplastic dynamics. We refer to this framework as GIOROM (**G**eometry-**I**nf**O**rmed **R**educed-**O**rder **M**odeling). All of our code and data is available at https://github.com/HrishikeshVish/GIOROM

## 1 Introduction

Various physical simulations involve simulating the spatio-temporal evolution of continuous fields governed by partial differential equations (PDEs) of the form

$$\boldsymbol{\mathcal{J}}(\boldsymbol{f}, \boldsymbol{\nabla}\boldsymbol{f}, \boldsymbol{\nabla}^2\boldsymbol{f}, \ldots, \dot{\boldsymbol{f}}, \ddot{\boldsymbol{f}}, \ldots) = \boldsymbol{0}, \tag{1}$$

$$\boldsymbol{f}(\boldsymbol{X}, t) : \Omega \times \mathcal{T} \to \mathbb{R}^d \tag{2}$$

where $\boldsymbol{f}$ represents a multidimensional continuous vector field that depends on both space and time (e.g., position field, velocity field, etc.). The symbols $\boldsymbol{\nabla}$ and $\dot{(\cdot)}$ signify the spatial gradient and time derivative, respectively. Here, $\Omega \subset \mathbb{R}^d$ and $\mathcal{T} \subset \mathbb{R}$ denote the spatial and temporal domains, respectively. To solve such a system, we define the solution operator $\boldsymbol{\mathcal{J}} : \boldsymbol{a} \to \boldsymbol{f}$, that maps a function of known observations $\boldsymbol{a} : \Omega' \times \mathcal{T}' \to \mathbb{R}^{d'}$ to $\boldsymbol{f}$, where we assume $\boldsymbol{a}$ and $\boldsymbol{f}$ to lie on Banach function spaces defined on bounded domains $D' \in \mathbb{R}^{d'}$ and $D \in \mathbb{R}^d$ respectively. We let $\mathcal{M}$ denote the manifold of solutions over all parameters and time and $d$ denotes the dimension (2 or 3).

To numerically solve the equation, the system is discretized spatially and temporally. Spatial discretization allows for the approximations $\boldsymbol{f}(\boldsymbol{X}, t) \approx \boldsymbol{f}_P(\boldsymbol{X}, t)$, and, $\boldsymbol{a}(\boldsymbol{X}, t) \approx \boldsymbol{a}_P(\boldsymbol{X}, t)$, where $P$ represents the $P$-point spatial discretization in $\mathbb{R}^d$, transforming $\boldsymbol{X}$ to a $(P \cdot d)$-dimensional vector. Similarly, we introduce temporal samples $\{t_n\}_{n=1}^T$, so that, for a given sample $x \in \{x^i\}_{i=1}^P \in \mathbb{R}^{P \cdot d}$, we can compute the spatiotemporal evolution using a set of input state variables $\boldsymbol{a}_t = \boldsymbol{a}_P(x, t)$ to obtain the solution $\boldsymbol{f}_{t+1} = \boldsymbol{f}_P(x, t+1)$ at the next state.

In this work, we focus on **particle interaction dynamics** modeled using Lagrangian systems Zefran & Bullo (2005), in which the temporal evolution of interacting particles (represented as point-clouds) is governed by equations of motion derived from a Lagrangian functional encoding kinetic and potential energy contributions.

It has been shown that the computational cost of solving $\boldsymbol{\mathcal{J}}$ scales with the resolution $P$, making it computationally expensive to use full-order PDE solvers, which operate directly on $P \cdot d$ degrees of freedom Chen et al. (2023; 2021).

Reduced order modeling approaches are used to overcome this computational bottleneck. Traditional ROMs achieve computational speedup by projecting high dimensional state variables onto low dimensional latent subspaces, typically via methods like Proper Orthogonal Decomposition (POD) or reduced basis techniques. This enables approximation of these dynamics on fewer degrees of freedom (e.g. $Q \ll P \cdot d$), reducing the

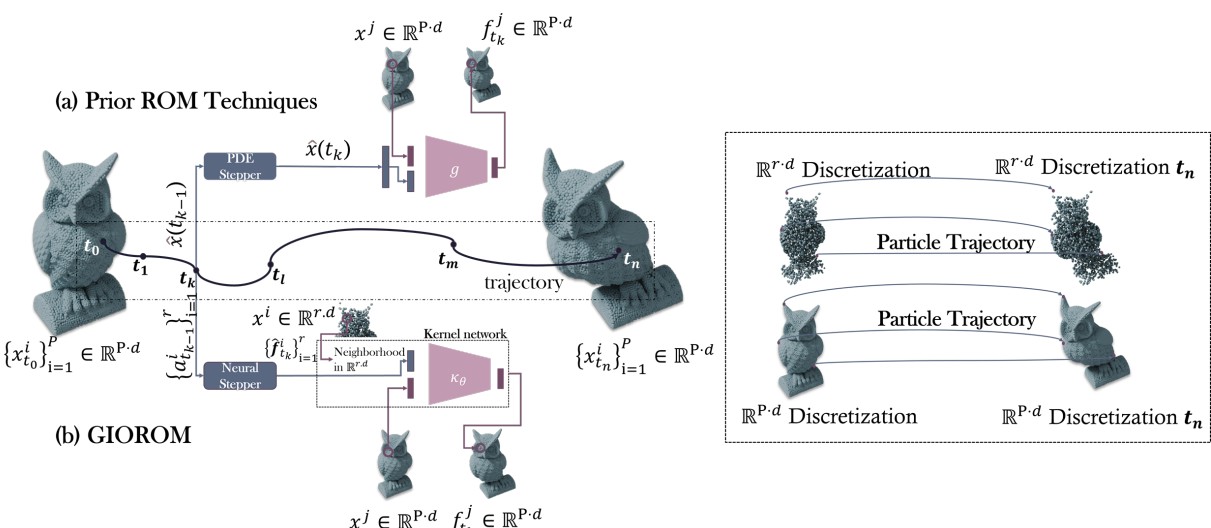

Figure 2: **Overview of GIOROM.** This figure illustrates the temporal evolution of Lagrangian interaction systems using reduced-order modeling. (a) Prior discretization invariant methods evolve a global latent state $\hat{\mathbf{x}}$ using PDE time-steppers and reconstruct the full-field solution using neural decoders $g$. This means, they require explicit PDE forms to compute the temporal evolution (b) In contrast, our approach employs a data-driven neural stepper to compute sparse field estimates $\{\hat{f}^i\}_{i=1}^r \in \mathbb{R}^{r \cdot d}$, which are then used by the kernel $\kappa$ to parameterize the solution manifold. In this case, the model does not require explicit PDE form. $f$ represents the deformation field depicted above. While the decoder processes the latent vector representing the entire spatial field, the kernel network evaluates over local spatial regions.

computational cost Lucia et al. (2004). These approaches operate under an offline-online paradigm: the expensive offline phase constructs a global basis from full-order simulations, and the inexpensive online phase evolves low-dimensional coefficients spatio-temporally using reduced forms of governing equations. To recover the solution field from the evolved latents, modern ROM approaches typically leverage continuous implicit neural representations such as neural fields. While effective in many regimes, these global latent representations often fail to capture complex, fast-evolving local dynamics within dynamic systems such as fluids, because of their lack of locality. Furthermore these approaches are not fully data-driven. They still rely on access to the underlying governing equations for spatiotemporal evolution.

To address this, we propose a new ROM framework that replaces projection-based reduction with a more direct, **sampling-based reduction** that enables reduced evolution directly in physical space. Specifically, we evolve the spatio-temporal dynamics of a small set of representative particles (samples) embedded in the physical domain, bypassing the need for a global latent state. This evolution can be facilitated using data-driven neural Lagrangian solvers Sanchez-Gonzalez et al. (2020), reducing dependencies on explicit forms of governing equations.

Furthermore, this direct "particle-space" evolution preserves locality while also reducing computational complexity by minimizing the number of active degrees of freedom (i.e. number of particles processed) during the evolution. This allows for a distinct advantage in constructing the solution manifold: discretization invariant kernel-based manifold parameterization can be defined over the evolved particle sample set to query arbitrary points, thus retaining the modular approach of traditional ROMs without sacrificing the locality of the physical space. Crucially, unlike traditional ROMs, which perform temporal evolution in reduced space, we follow a hybrid approach of evolving the Lagrangian particles in sampled space, while leveraging kernel based manifold parameterization as a reduced spatial representation to parameterize the high resolution spatial field. The temporal evolution is achieved with the discretization agnostic neural operator.

The framework remains **fully agnostic** to the underlying neural operator architecture, enabling it to exploit the strengths of established neural physics models that accurately model particle dynamics and generalize

across PDE parameterizations. Through reduced sampling of particle sets, GIOROM further enhances these models, leveraging their robustness to discretization variations to improve their inference speeds.

Thus, our proposed GIOROM framework introduces a fully data driven ROM framework that retains essential ROM characteristics: modularity, reduced-order evolution, and continuous implicit spatial parameterization of the solution field, while performing time-stepping directly in physical space. The formulation remains independent of the underlying discretization.

## 2 Related Works

**Reduced-order models** ROMs reduce computational cost by projecting high-dimensional systems onto low-dimensional manifolds Berkooz et al. (1993); Holmes et al. (2012); Lee & Carlberg (2020); Peherstorfer (2022), often via sample selection An et al. (2008). These low dimensional states are explicitly and **independently** evolved using time-steppers. Neural field-based ROMs Pan et al. (2023); Yin et al. (2023); Wen et al. (2023); Chen et al. (2023); Chang et al. (2023) enable discretization-agnostic learning, supporting generalization across geometric discretizations, but are still intrusive, requiring exact PDE formulation to time-step. Data-driven reduced-order modeling use deep-learning based techniques to achieve order-reduction Guo & Hesthaven (2019); Peherstorfer & Willcox (2015); Besabe et al. (2025), but these are however, not discretization invariant and require retraining to adapt to different discretizations. Kernel-based ROM techniques Honeine (2011); Kou & Zhang (2017); Raveh (2001); Munteanu et al. (2005); Salvador et al. (2021) leverage kernel methods such as Kernel-PCA or Kernel-POD, but they are discretization-dependent. Additional works are in section B. Our work can be seen as discretization invariant kernel-integral ROM parameterization for reduced space that is both discretization invariant and non-intrusive.

**Neural physics solvers** Neural solvers have advanced simulations across fluid dynamics Sanchez-Gonzalez et al. (2020); Kochkov et al. (2021); Vinuesa & Brunton (2021); Mao et al. (2020); Shukla et al. (2024); Hao et al. (2024), solid mechanics Geist & Trimpe (2021); Capuano & Rimoli (2019); Jin et al. (2023), climate modeling Pathak et al. (2022), and robotics Ni & Qureshi (2022); Kaczmarski et al. (2023). These models range from data-driven to physics-informed architectures Raissi et al. (2019); Sirignano & Spiliopoulos (2018); Richter & Berner (2022); Nam et al. (2024). CNNs are effective on regular grids Lee & Carlberg (2020); Maulik et al. (2021); Stoffel et al. (2020); Bamer et al. (2021), while GNNs generalize to irregular meshes, enabling applications in mesh-based dynamics Pfaff et al. (2020); Cao et al. (2022); Han et al. (2022); Fortunato et al. (2022), Lagrangian systems Sanchez-Gonzalez et al. (2020), parametric PDEs Pichi et al. (2024), and rigid body physics Kneifl et al. (2024). Lagrangian modeling, which depends on particle-interactions are typically modeled with GNN based message passing. However, GNNs such as GNS and Meshgraphnets scale poorly with graph size due to node-wise message passing defined over all degrees of freedom. While traditional ROMs parameterize latent-spaces, our proposed reduction strategy can leverage these models directly, while achieving order reduction.

**Neural operators** Neural operators are a class of discretization-invariant models for learning mappings between infinite-dimensional function spaces. They have been applied to parametric PDEs Lu et al. (2021); Li et al. (2020c; 2023); Azizzadenesheli et al. (2024); Rahman et al. (2024); Liu-Schiaffini et al. (2024); Kovachki et al. (2023); Rahman et al. (2022a); Liu et al. (2022); Viswanath et al. (2023); Shih et al. (2024); Goswami et al. (2023), fluid simulations Di Leoni et al. (2023); Wang et al. (2024); Peyvan et al. (2024), 3D physics Xu et al. (2024); White et al. (2023); Bonev et al. (2023); Rahman et al. (2022b); He et al. (2024), and even cross-domain tasks in biology, robotics, and vision Liu et al. (2024a); Dharuman et al. (2023); Bhaskara et al. (2023); Peng et al. (2023); Guibas et al. (2021); Rahman & Yeh (2024); Viswanath et al. (2022). While operators such as FNOs, and Transolver Wu et al. (2024a), use projections within their frameworks, they learn a direct end-to-end function mapping rather than evolving a reduced state. Their architectures couple the projection and processing steps, meaning the online evaluation phase still depends on the full-order degrees of freedom of the input. However, DeepONet Lu et al. (2021) can be viewed as a form of Eulerian ROM solver. On the other hand, our work, bridges the gap between neural Reduced Order Models (ROMs), neural Lagrangian solvers and neural operators. We propose a principled framework for integrating operator learning within order reduction paradigm.

Our proposed reduction strategy can be used in Geometric Informed Neural Operator (GINO) Li et al. (2024) and the Unified Physics Transformer (UPT) Alkin et al. (2024) to reduce effective degrees of freedom, however, GINO is unsuitable for particle-interactions. The UPT family of architectures follow the encode-process-decode framework, with graph based layers facilitating Lagrangian inputs. But their architectures evaluate the solution field, rather than particle interactions and remain unsuitable for autoregressive particle dynamics.

Table 1: **Contrasting neural PDE operators and neural ROM solvers**: The upper half of the table represent features associated with neural physics solvers while the lower half represents features in neural ROM solvers

| Category | FNO | GNS / MeshgraphNet | Transolver | GINO | UPT | DINo | CROM / LICROM | Ours |
|---|---|---|---|---|---|---|---|---|
| Discretization Invariance | ✔[1] | ✔ | ✔ | ✔ | ✔ | ✔ | ✔ | ✔ |
| Irregular Domain Support | ✗ | ✔ | ✔ | ✔ | ✔ | ✔ | ✔ | ✔ |
| Neural PDE Operator | ✔ | ✔ | ✔ | ✔ | ✔ | ✗ | ✗ | ✔ |
| Locality Preserving Reduced Space | ✗ | ✗ | ✗ | ✔ | ✗ | ✗ | ✗ | ✔ |
| Particle Interaction Dynamics | ✗ | ✔ | ✗[5] | ✗ | ✗[6] | ✗ | ✔ | ✔ |
| Low-Dim State Representation | ✔[2] | ✗ | ✔ | ✔ | ✔ | ✔ | ✔ | ✔ |
| Decoupled Latent Space Dynamics | ✗ | ✗ | ✗ | ✗ | ✔ | ✔ | ✔ | ✔ |
| Autoregressive Reduced Evolution | ✗ | ✗ | ✗ | ✗ | ✗[6] | ✔ | ✔ | ✔ |
| Modularity | ✗ | ✗ | ✗ | ✗ | ✔ | ✔ | ✔ | ✔ |
| Effective Active DOFs (Online) | $N$ | $N$ | $N$ | $^4r \ll N$ | $^4r \ll N$ | $r \ll N$ | $r \ll N$ | $r \ll N$ |
| Non-Intrusive | ✔ | ✔ | ✔ | ✔ | ✔ | Partially[3] | ✗ | ✔ |

**Notes:**

[1] **FNO:** While designed to be discretization-invariant, standard implementations perform best on regular grids and may require adaptation for irregular meshes.

[2] **FNO:** The "reduction" is in the learned operator, which is parameterized by a small number of Fourier modes, serving as the effective latent space, but not decoupled from the projection operator.

[3] **DINo:** Can be considered "partially intrusive" as they are designed around a neural ODE formulation governing the latent space. DINo considers Initial Value Problems (IVP) where the trajectories follow the same dynamics but have different initial conditions.

[4] **GINO/UPT:** Due to their ability to extrapolate inputs over arbitrary query points, their active degrees of freedom can be effectively reduced to a sparse set of points using our proposed reduction strategy.

[5] **Transolver:** The physics attention mechanism of Transolver does not support Lagrangian inputs, but can be combined with GNN based pre-processing layers to handle Lagrangian systems.

[6] **UPT:** The UPT architecture models field characteristics of Lagrangian systems instead of tracking individual particles and is therefore unable to model autoregressive evolution, wherein the encoder requires particle positions as input.

# 3 Method: Kernel parameterization

We formulate the reduced-order model (ROM) not as a projection onto a fixed global basis, but as a continuous integral operator parameterized by an implicit kernel. We refer to this as kernel-integral ROM. Let $\mathcal{H} = L^2(\Omega; \mathbb{R}^k)$ denote a Hilbert space of state functions over a bounded domain $\Omega \subset \mathbb{R}^d$, and assume the solution $\boldsymbol{f}(\cdot, t)$ evolves on a low-dimensional manifold embedded in $\mathcal{H}$. $\boldsymbol{f}$ is assumed to lie on a continuous, smooth solution manifold $\mathcal{M}$. We do not make assumptions regarding access to its closed form and instead are provided with a set of pointwise approximations $\{x^i, \hat{\boldsymbol{f}}^i\}_{i=1}^r$, where $r$ is the number of sparse samples.

To recover the continuous field structure from the sparse samples assumed to lie on the smooth solution manifold, we introduce a learnable kernel parameterization $\psi(x, y)$ that allows the field value at a query point $x$ to be reconstructed as a weighted aggregation of nearby Lagrangian samples. Since $\psi$ is not assumed to define a probability density, the reconstruction is expressed as a normalized integral operator, enforcing a partition-of-unity-like condition and ensuring consistency under uniform fields. This yields a nonlinear, location-dependent operator that can be viewed as a continuous analogue of kernel regression, closely related

to Nadaraya–Watson estimators Cai (2001) and graph-based kernel integral transforms. Unlike global projection-based ROMs, this formulation induces a spatially adaptive operator whose effective support and conditioning vary with the local sample geometry. Consequently, the particle features within the local support of a query point act as the degrees of freedom (DoF) for field evaluation, without the explicit orthonormality constraints required by global basis projections.

For a query point $x \in \Omega$, the reconstructed value is formally given by the ratio of expectations:

$$u(x,t) = \frac{\int_{\Omega} \psi(x,y)\, v(y,t)\, d\mu(y)}{\int_{\Omega} \psi(x,y)\, d\mu(y)}, \tag{3}$$

where $\psi$ is a smoothing kernel and the denominator ensures density normalization, $v \approx \boldsymbol{f}$ denotes the known Lagrangian sample field, obtained via the PDE operator (i.e. time-stepper) $\phi_{\Theta}$ (discussed in section 4) and $d\mu$ represents the weighted Lebesgue measure on $\Omega$.

This formulation is similar to graph based kernel integral transform (as in GINO Li et al. (2024)), however, direct evaluation via radius-based graphs scales with the sample size $r$ and is computationally expensive for high-resolution Lagrangian simulations. To mitigate this, we adopt a *projection–stochastic sampling strategy.*

The Lagrangian samples are projected onto a fixed grid using trilinear basis functions $\phi_h$, producing a discrete density field $D_{grid}$ and a feature field $F_{grid}$ defined at the grid nodes $\mathbf{g}$: $F_{grid}(\mathbf{g}) = \sum_j \phi_h(\mathbf{g}, y_j) v(y_j, t)$, $D_{grid}(\mathbf{g}) = \sum_j \phi_h(\mathbf{g}, y_j)$. The local integral operator is then approximated via stochastic interpolation of these grid fields around the query location $x$:

$$\boldsymbol{f}(x,t) \approx \bar{u}(x) \;=\; \frac{\mathbb{E}_{\xi}\left[\mathcal{I}(F_{grid}, x + \xi)\right]}{\mathbb{E}_{\xi}\left[\mathcal{I}(D_{grid}, x + \xi)\right] + \epsilon}, \tag{4}$$

where $\mathcal{I}(\cdot, \cdot)$ denotes trilinear interpolation, $\xi \sim \mathcal{N}(0, \sigma^2 I)$ is a stochastic perturbation, and $\epsilon$ is a small regularization for numerical stability. The stochastic perturbation $\xi$ induces a localized averaging that acts as a Monte Carlo approximation of the convolution with a Gaussian smoothing kernel. This avoids the explicit construction of radius-based neighborhoods or dense graph connectivity. Similar to the radius graphs, this step is discretization convergent on the sample size $r$. We observe empirically that this formulation achieves computational improvements over radius based aggregation while achieving better performance. We discuss the exact implementation details of the parameterization in section H.2.

**Loss Objective** The training objective for the learned field $\bar{u}(x)$ is given by:

$$\mathcal{L}_{\theta} = \mathbb{E}_t\left[\sum_{i=1}^{P} \|\bar{u}(x_i, t) - u_{GT}(x_i, t)\|_2^2\right]. \tag{5}$$

## 4 Method: Time-stepping on reduced sample set

We make a deviation in this section to focus on obtaining the sample set $\{x^i, \hat{\boldsymbol{f}}^i\}_{i=1}^r$, which is done using the operator $\phi_{\Theta}$. With $r \ll P$, we only require evaluating the *full-order dynamics* on a sparse set of points, achieving computational speedups. Following the UPT family of transformer architectures Alkin et al. (2024), we adapt the UPT backbone to enable autoregressive particle dynamics using the interaction network based encoder and decoder Battaglia et al. (2016). The interaction modules allow the operator framework to capture interaction dynamics and can be used in autoregressive generations akin to Sanchez-Gonzalez et al. (2020). However, the use of transformer processor enables speedups over GNS and MeshgraphNet style architectures. The modular nature of the architecture ensures the operator is agnostic to the choice of the transformer and is compatible with any PDE operator transformers such as Hao et al. (2023); Wu et al. (2024a); Lee & Oh (2024); Alkin et al. (2024).

### 4.1 Step 1: Input representation

We consider a reduced set of material points $\{x^i\}_{i=1}^r$, with $r \ll P$, for which we seek full-order evaluations without assuming a specific sampling scheme. Importantly, we do not train $\phi_\Theta$ for a fixed $r$. The model $\phi_\Theta$ is trained on coarse discretizations $\{x^i, a^i, \boldsymbol{f}^i\}_{i=1}^{|\mathbf{V}|}$, where $|\mathbf{V}| \ll P \cdot d$, defined on a radius graph $\mathbf{G} = (\mathbf{V}, \mathbf{E})$. Here, $\mathbf{V}$ denotes particles and $\mathbf{E}$ contains undirected edges between nodes within radius $\rho_s$, i.e., $(x, y) \in \mathbf{E} \iff d(x, y) \leq \rho_s$. The edges represent particle interactions.

For a subgraph $\mathbf{H} = (v, \varepsilon)$, defined over $v \subseteq \mathbf{V}$ with edges induced by radius $\rho_r$, we view $\mathbf{G}$ and $\mathbf{H}$ as discrete approximations of an underlying continuous manifold $\mathcal{C} \subset \mathbb{R}^d$ Jin et al. (2020); Burago et al. (2015). We observe that for reasonable sizes of $\mathbf{H}$, we can tune the radius $\rho_r$ and enforce that for $x^i \in \mathbf{V} \cap v$, $\phi_{\Theta, \mathbf{H}}(x^i, t) \approx \phi_{\Theta, \mathbf{G}}(x^i, t) := \hat{\boldsymbol{f}}^i$, where $\phi_{\Theta, \mathbf{G}}$ represents the model operating on graph $\mathbf{G}$. This is achieved by enforcing a discrete neural operator behavior within the GNN architecture. We require tuning of $\rho_r$ as fewer nodes alter local neighborhoods, which can disrupt message passing Garg et al. (2020); Gao & Isufi (2022).

### 4.2 Step 2: Heuristic for graph construction

For a subgraph $\mathbf{H}$ defined on $v \subseteq \mathbf{V}$, we require the connectivity radius $\rho_r \geq \rho_s$ to preserve paths between originally connected nodes, even after subsampling; $r$ is chosen to optimize the kernel network performance/cost tradeoff, and consequently, $\rho_r$ is tuned at inference to trade off accuracy and speed.

We use mean aggregation in message passing to ensure consistent node features across discretizations. This is shown to behave like a Monte Carlo approximation of kernel integrals, enabling the GNN to function as a neural operator under suitable graph construction Li et al. (2020b).

Prior sampling-based ROM approaches rely on stochastic, residual-guided strategies to select sample sets Chen et al. (2023). In contrast, we adopt simple sampling methods such as random or farthest-point selection. To ensure invariance to the sampling ratio for $r := |v|/|\mathbf{V}|$, we construct $l$ random subgraphs per input domain to evaluate during training. This can be seen as a Nyström-type approximation, reducing variance and improving generalization across discretizations.

Empirically, we find that overly small $|v|$ degrades accuracy even with large $\rho_r$, implying a lower bound on $r$ to retain spatial structure. This threshold guides inference-time subgraph construction (section M). Conversely, excessive $\rho_r$ introduces redundant edges and deteriorates performance due to loss of locality.

### 4.3 Step 3: Spatio-temporal time-stepping

Equipped with our data structure $\mathbf{H}$, we now discuss the architecture of $\phi_\Theta$ that enables the mapping $a_t^j \to \hat{\boldsymbol{f}}_{t+1}^j$. We define $a^j$ and $f^j$ to be point-wise function values defined on the node set $v \in \mathbb{R}^{r \cdot d}$.

Extending the UPT family of architectures, our formulation can then be defined as $\phi_\Theta := IN^{dec, \mathbf{H}} \circ NOT^{process} \circ IN^{enc, \mathbf{H}}$, where $enc$ and $dec$ denote the encoder and decoder, which operate on the graph $\mathbf{H}$. The NOT however operates only on the latent embeddings. The input for the model is the velocity sequence $\mathbf{v}$ defined over a time window $w$. Formally, $a_t^j = \mathbf{v}_{t-w:t}^j, \forall j \in \mathbb{R}^{r \cdot d}$. The output is then given as $\ddot{\hat{\boldsymbol{f}}}_t^j = \mathbf{a}_t^j$, where $\mathbf{a}$ is the pointwise acceleration defined on $v$. We obtain $\hat{\boldsymbol{f}}_{t+1}^j$ by Euler integration. The model is trained following Sanchez-Gonzalez et al. (2020) with data-driven losses as follows $\mathcal{L}_\Theta = \min_\Theta \sum_j \|\phi_\Theta(a_t^j) - \ddot{\boldsymbol{f}}_t^j\|_2^2$ Additional training details and hyperparameter information is provided in section H and section J

## 5 Dataset

Our dataset consists of four 3D physical systems - Newtonian fluids (Water), Drucker-Prager elastoplasticity (Sand), von Mises yield (Plasticine) and purely Elastic deformations. Unless otherwise noted, we assume the discretization to be point clouds.

We used the nclaw simulator (Ma et al., 2023) to generate 100 trajectories for each of these systems with random initial velocity conditions with a free-slip boundary condition, typically used in MPM simulations. The

$\Delta t$ between consecutive time frames is $5e^{-3}$s. We additionally used datasets provided by Sanchez-Gonzalez et al. (2020) - WaterDrop, Water-3D, Sand-3D, Sand, Goop and MultiMaterial. Additional training details and model hyperparameters can be found in Appendix H.2.

## 6 Experiments

This section is structured to evaluate the two core contributions of GIOROM: the **locality of the kernel-integral ROM formulation** (space) and the **computational efficiency of the PDE agnostic dynamics** (time).

**Manifold parameterization and generalization (space)** We first investigate the fidelity of the ROM basis functions by benchmarking our kernel formulation against prior ROM basis representations. We categorize these baselines based on their latent space design:

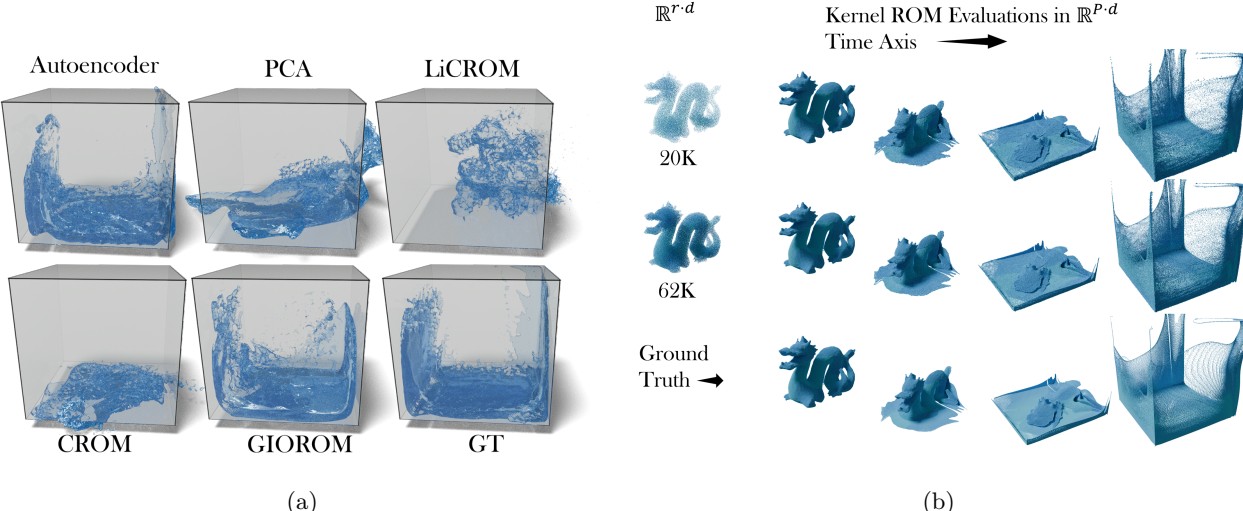

(a)                  (b)

Figure 3: **(a) Comparison of reduced-order modeling (ROM) techniques on fluid-in-container simulations**: The visualization reveals that the baseline models PCA (rollout MSE: 0.083), Autoencoder (0.091), LiCROM (0.033), and CROM (0.079), exhibit noticeable deviation from the true fluid boundaries, including overshooting beyond the container. GIOROM maintains physical fidelity with a lower rollout MSE of 0.0091. (b) **Demonstration of discretization convergence of kernel-integral ROM on high-resolution particle systems**: Contrasting the behavior of the kernel-integral ROM on a large-scale simulation containing approximately **2 million particles**, discretized as voxelized grids. Using randomly sampled $(r \cdot d)$-points of 62K and 20K particles (sampling percentages of 3% and 1%), the rollout MSEs are 0.0097 and 0.0088, respectively. These results indicate discretization convergence of $\kappa$ under resolution refinement.

*Continuous Space INR ROMs.* These baselines represent architectures that traditionally rely on explicit PDE forms to evolve the latent space dynamics. We evaluate the global bases utilized in CROM Chen et al. (2023), CoLORA Berman & Peherstorfer (2024), and LiCROM, alongside the local graph kernel integral transform (GKI) from GINO Li et al. (2024). While Transolver, FNO and GINO include projection layers, they are trained end-to-end in operator setting, but the integral transform (GKI) serves as a discretization-convergent kernel formulation that operates directly in sample space, forming a key baseline for local parameterization methods. To evaluate the spatial expressivity of these manifolds, we employ a common time-stepper to evolve the sample space trajectory for all static ROMs. The full-order state is subsequently reconstructed via the respective basis encoders. This allows us to compare spatial parameterizations of the baseline projection ROMs and overcome their PDE dependencies for temporal rollouts.

*Continuous time latent dynamics ROMs.* We further evaluate architectures such as DINo Yin et al. (2023) and CORAL Serrano et al. (2023), which integrate non-linear autoencoders with intrinsic Neural ODEs. For these models, we train the intrinsic Neural ODE to evolve the latent states temporally. Reconstruction metrics are then computed on the decoded full-order states.

**End-to-end computational efficiency (time)**   Following the manifold validation, we evaluate the end-to-end performance of the full GIOROM pipeline against GNS Sanchez-Gonzalez et al. (2020), the state-of-the-art full-order Lagrangian graph solver. In this context, GNS serves as a high-fidelity oracle for rollout accuracy. These experiments demonstrate that our latent time-stepping on the reduced manifold achieves rollout stability comparable to the full-order solver.

## 6.1   Results: Manifold parameterization

**Generalization**   We showcase quantitative analyses in table 2. We observe that GIOROM achieves the lowest Relative $L_2$ error and Chamfer distance in the four physical systems on **previously unseen trajectories (i.e. initial conditions)**. The performance gap is most pronounced in the `WATER-3D` dataset, a regime characterized by topological breaks and highly dynamic transitions. The learnable latent dynamics baselines (DINo, CORAL) exhibit high variance and error rates ($> 30\%$) in these multi-trajectory settings, whereas the continuous space INR ROMs exhibit better performance. We additionally present the results on `WATER-3D` in fig. 3a. The baselines we compared include (1) PCA (Principal Component Analysis), also known as POD (Proper Orthogonal Decomposition), (2) Autoencoders Lee & Carlberg (2020), (3) LICROM Chang et al. (2023), and (4) CROM Chen et al. (2023).

**Computational footprint**   We further analyze the trade-off between performance and computational cost. While PCA is trivially the fastest ($\approx 0.5$ ms) with negligible memory, it fails to capture non-linear deformations, resulting in high Chamfer distances. Among the non-linear deep learning methods, the Jax based GIOROM offers the most favorable resource-to-accuracy ratio. It requires an order of magnitude less peak memory (17–267 MB) compared to the graph-kernel baseline GKI (462–2970 MB) and the latent ODEs, while operating at inference speeds (0.79–5.47 ms) that approach the linear baseline.

**Impact of reduced space**   To quantify the manifold capacity across ROM architectures, we analyze error convergence with respect to the reduced space size. For global ROMs (e.g., PCA, CROM), this corresponds strictly to the fixed latent dimension $Q$. In contrast, GIOROM does not project the global state into a single latent vector; instead, it distributes information onto a sparse grid. To compare these architectures, we view each active grid cell as a local reference vector for the query points within its support. Accordingly, we define a heuristic for reduced size $\mathcal{S}_{\text{eff}}$ as the ratio of total Lagrangian feature size to the active grid voxels: $\mathcal{S}_{\text{eff}} \approx N_p \times C / N_{\text{active}}$, where $N_p$ is the Lagrangian sample size and $C$ is the feature size. This represents heuristic for the expected number of features that contribute to each active cell. This framework represents a **relaxation** of the strict projection-based ROM paradigm, where time-stepping is confined entirely to the reduced latent space. In our hybrid formulation, we decouple these complexities: the **spatial parameterization** is governed by the sparse grid in equation 4, while the **time-stepping** is defined over the Lagrangian samples.

Figure 6 illustrates the convergence behavior under this metric. It can be seen that in the case of topologically stable regimes such as `PLASTICINE-3D` and `ELASTICITY-3D`, GIOROM performs similarly to global bases. This is attributed to the preservation of relative particle positioning, which allows global linear bases to effectively capture smooth deformations. However, we observe that global bases outperform GIOROM in extremely sparse regimes. This occurs because in such settings, each active voxel is effectively influenced by a single particle. In contrast, global ROMs utilize learned basis functions (e.g., principal components) making them more expressive. We present visual comparisons for different regimes in fig. 17, fig. 18, fig. 19, and fig. 20.

**Computational scalability**   To empirically verify the computational footprint in section 3, we benchmark GIOROM against the Graph based GKI baseline Li et al. (2024). We analyze scalability with respect to two factors: the interaction radius $\delta$ and the sample space particle count $r$.

*Sensitivity to interaction radius (memory scaling)* We compared the memory footprint of the neighborhood graph structure (GKI) versus our grid-projection hash (GIOROM) for a fixed particle set ($N = 20,000$) while varying the normalized smoothing radius $\delta$. As shown in fig. 4.a, GKI exhibits explosive memory growth, scaling from 2.8 MB at $\delta = 0.01$ to 1.37 GB at $\delta = 0.2$. In contrast, GIOROM maintains a constant 7.0 MB memory footprint regardless of $\delta$.

*Scaling with sparsity ($N$)* We further decompose the wall-clock inference time into "Structure Build" (Graph Search vs. Grid Projection) and "Function Evaluation" for particle counts ranging from $N = 2,000$ to $35,000$. This is shown in fig. 4.b and fig. 4.c. Our results reveal that for GKI, the graph construction is the dominant bottleneck, accounting for 97% of the total runtime at $N = 35k$ (176.9 ms build vs. 5.1 ms compute). GIOROM effectively eliminates this bottleneck. Furthermore, while GKI memory scales linearly with $N$ (reaching 83 MB at $N = 35k$), GIOROM's structure memory remains fixed at 7 MB, determined by the grid resolution. Lastly, as the grid is discretization convergent, it exhibits degradation with extreme reduction in particle counts. We analyze this in section M.2 and provide additional ablations on kernel parameters.

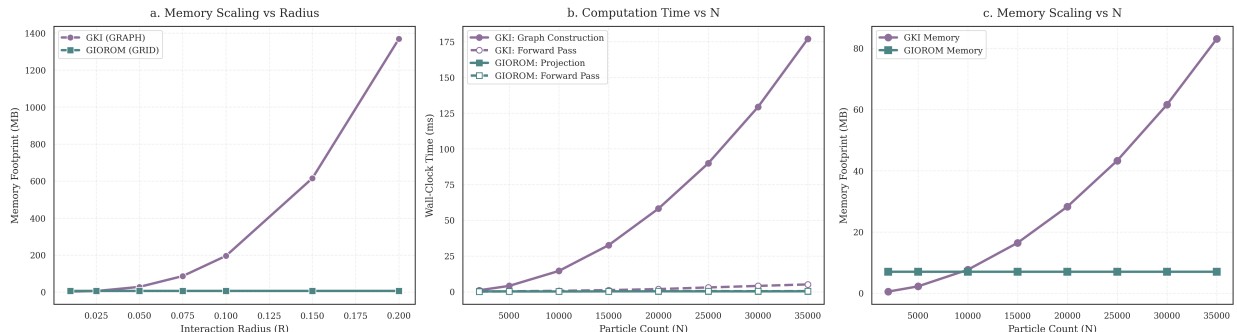

Figure 4: **Computation overheads in implicit kernel based ROMs** These plots showcase the overheads in performing radius based graph integral transform, with plot (a) highlighting the scaling as the radius increases, plots (b) and (c) highlighting the time and memory overheads as the particle count scales. In contrast, grid projection maintains fairly stable computation overhead.

## 6.2 Results: End-to-end dynamics and scalability

**Comparison with full-order solvers.** We first evaluate the ability of the reduced model to preserve dynamic stability compared to the full-order baseline. table 3 reports the one-step and rollout MSE of GIOROM against the GNS oracle across diverse physical systems. We show that operating on a sparsified graphs, reduced by factors of up to $32\times$ (e.g., `WATER-3D`, `SAND-3D`). The error margins remain within the same order of magnitude as the GNS baseline, even for complex multi-material interactions. Additional results and discussions are provided in section K and section M.

**Computational improvements** While Graph Neural Networks (GNNs) effectively capture spatial interactions in Lagrangian systems, the iterative message-passing operations introduce significant computational bottlenecks. This overhead scales poorly with edge density. GIOROM mitigates this by adopting a UPT-style Encoder-Process-Decoder architecture, where global temporal mixing is handled by a Transformer processor rather than recursive message passing on the graph edges. We evaluate this in two settings, first by increasing the number of edges by tuning the graph radius, and second by increasing the number of nodes by tuning the sampling rate.

*Graph Radius* table 4 isolates the impact of the connectivity radius on inference time. As the radius increases from 0.040 to 0.100, the inference time of GNS degrades drastically (from 43.5s to 386.0s). In contrast, GIOROM time-stepper exhibits far more robust scaling behavior, maintaining a speedup factor that widens from $\approx 2\times$ at low connectivity to $\approx 3.5\times$ at high connectivity (109.3s vs 386.0s).

Table 2: **Per-Dataset Quantitative Evaluation.** We report Rel. $L_2$, Chamfer, and VRAM. GIOROM achieves the best geometric fidelity with constant-time efficiency. **Bold** is best, underlined is second. All the results are computed on a reduced space with 32 effective active degrees of freedom

| DATASET | MODEL | REL $L_2$ ERROR (%) ↓ | CHAMFER ($L_2$) ($\times 10^{-4}$) ↓ | TIME (ms) ↓ | AVG MEMORY (MB) ↓ | PEAK MEMORY (MB) ↓ |
|---|---|---|---|---|---|---|
| PLASTICINE-3D | PCA | 16.15% ± 0.96% | 19.25 | **0.42** | **0.02** | **4.42** |
| | DINo | 31.83% ± 11.72% | 434.74 | 1.66 | 477.85 | 478.00 |
| | CORAL | 21.85% ± 8.46% | 141.81 | 2.71 | 407.87 | 408.00 |
| | CoLORA | 9.08% ± 3.90% | 21.05 | 1.25 | 407.87 | 408.00 |
| | CROM | 3.19% ± 0.62% | 4.58 | 1.49 | 407.87 | 408.00 |
| | LICROM | 2.31% ± 0.46% | 3.78 | 4.23 | 474.80 | 476.00 |
| | GKI | 3.69% ± 3.89% | 8.64 | 3.95 | 461.87 | 462.00 |
| | **GIOROM** (Ours) | **1.98% ± 0.57%** | **2.75** | 0.79 | 0.60 | 17.28 |
| SAND-3D | PCA | 13.37% ± 1.91% | 26.32 | **0.51** | **0.01** | **1.69** |
| | DINo | 45.36% ± 7.78% | 319.17 | 3.43 | 533.51 | 534.00 |
| | CORAL | 25.80% ± 8.06% | 66.44 | 3.57 | 407.62 | 408.00 |
| | CoLORA | 12.31% ± 4.31% | 23.56 | 1.59 | 407.62 | 408.00 |
| | CROM | 4.21% ± 0.56% | 6.61 | 1.55 | 407.62 | 408.00 |
| | LICROM | 3.44% ± 0.65% | 6.26 | 9.15 | 513.52 | 522.00 |
| | GKI | 8.46% ± 5.24% | 28.31 | 23.41 | 952.26 | 952.40 |
| | **GIOROM** (Ours) | **2.68% ± 0.91%** | **2.55** | 3.25 | 2.07 | 132.24 |
| WATER-3D | PCA | 15.84% ± 3.60% | 57.52 | **0.54** | **0.01** | **1.69** |
| | DINo | 58.97% ± 13.47% | 483.62 | 3.25 | 575.15 | 592.00 |
| | CORAL | 53.71% ± 15.16% | 323.06 | 4.15 | 428.61 | 430.00 |
| | CoLORA | 54.54% ± 12.43% | 439.11 | 2.77 | 429.62 | 430.00 |
| | CROM | 24.00% ± 5.75% | 138.02 | 2.12 | 428.85 | 430.00 |
| | LICROM | 22.01% ± 5.76% | 132.21 | 13.63 | 546.30 | 548.00 |
| | GKI | 15.16% ± 6.95% | 45.33 | 79.89 | 2970 | 2970 |
| | **GIOROM** (Ours) | **9.24% ± 4.13%** | **8.67** | 3.51 | 9.33 | 266.72 |
| ELASTICITY-3D | PCA | 39.41% ± 0.66% | 63.04 | **0.56** | **0.01** | **1.05** |
| | DINo | 41.18% ± 10.01% | 165.95 | 4.50 | 617.47 | 629.00 |
| | CORAL | 46.24% ± 12.17% | 242.50 | 5.09 | 441.60 | 442.00 |
| | CoLORA | 9.83% ± 3.90% | 10.10 | 3.66 | 441.60 | 442.00 |
| | CROM | 4.25% ± 0.91% | 5.93 | 3.33 | 441.60 | 442.00 |
| | LICROM | 3.43% ± 1.27% | 5.77 | 7.98 | 547.74 | 549.75 |
| | GKI | 7.66% ± 0.80% | 6.89 | 42.53 | 1788 | 1788 |
| | **GIOROM** (Ours) | **2.04% ± 0.49%** | **3.03** | 5.47 | 9.69 | 267.81 |

*Graph Nodes* table 5 highlights a critical distinction between number of nodes and runtime cost. GIOROM consistently achieves $2\times$–$2.5\times$ faster wall-clock inference across all graph sizes. For the largest system (7056 points), GIOROM completes the 200-step rollout in 47.6s compared to 111.6s for GNS.

**Discretization Convergence** We analyze the sensitivity of the learned operator to spatial discretization using the `ELASTICITY-3D` system (78k particles). As illustrated in fig. 5, the rollout error remains remarkably stable (MSE $\approx 10^{-4}$) across sparsity levels ranging from $0.125\times$ down to $0.031\times$, confirming mesh-independent physics learning up to a $32\times$ compression ratio. Performance degrades only below this threshold due to over-sparsification; notably, in extreme regimes ($0.007\times$), increasing the connectivity radius beyond 0.15 yields diminishing returns, indicating that long-range edges cannot compensate for lost local geometry. Computationally, the method exhibits linear scaling in wall-clock time and peak GPU usage (inclusive of dynamic graph construction overhead) as the system is sparsified, ensuring that the cost of rebuilding the latent topology does not negate the speedups from dimensionality reduction.

## 7 Discussions and Conclusion

**Adaptive Time Discretization.** While our current implementation utilizes a fixed-step latent solver, the continuous manifold formulation is naturally compatible with adaptive temporal integration. A future direction could involve exploring the continuous time dynamics as used in DINo and CORAL, such as with a Neural ODE (NODE) allowing, the solver to dynamically adjust computational effort during periods of complex dynamics.

Table 3: Comparative performance of GIOROM (Ours) against the GNS baseline across diverse physical systems. Our model operates on a compressed latent graph of size $r$, where **Scale** ($N/r$) represents the reduction in graph complexity compared to GNS. All error metrics are computed on the full particle set $\mathbb{R}^{N \cdot d}$.

| PHYSICAL SYSTEM | FULL SIZE ($N$) | LATENT ($r$) | SCALE | ONE STEP-MSE ($\times e^{-9}$) Ours | GNS | ROLLOUT MSE ($\times e^{-3}$) Ours | GNS |
|---|---|---|---|---|---|---|---|
| WATER-3D | 55k | 1.7k | 32× | 5.23 | 8.78 | 38.6 | 17.4 |
| WATER-2D | 1k | 0.12k | 8.3× | 0.524 | 1.31 | 6.70 | 7.55 |
| SAND-3D | 32k | 1k | 32× | 4.87 | 3.38 | 0.25 | 0.31 |
| SAND-2D | 2k | 0.3k | 6.6× | 8.50 | 6.50 | 1.34 | 1.21 |
| GOOP-2D | 1.9k | 0.2k | 9.5× | 1.31 | 3.02 | 0.94 | 0.55 |
| PLASTICINE | 5k | 1.1k | 4.5× | 0.974 | 1.02 | 0.50 | 0.71 |
| ELASTICITY | 78k | 2.6k | 30× | 0.507 | 0.38 | 0.20 | 0.3 |
| MULTI-MAT 2D | 2k | 0.25k | 8× | 2.30 | 1.99 | 9.43 | 12.1 |

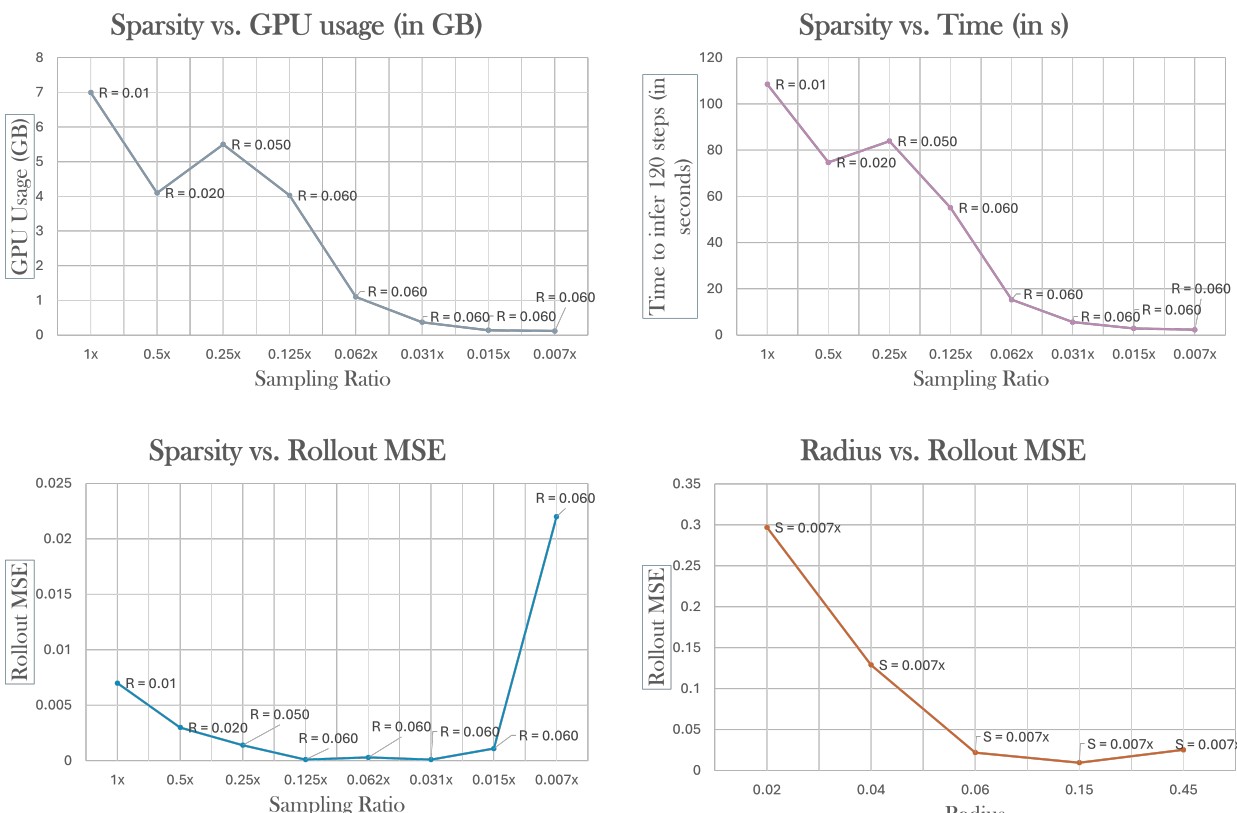

Figure 5: **Effect of sparsity on rollout error (Elasticity dataset, 78K particles).** Plots show performance of $\phi_\Theta$ across varying sparsity levels, expressed as a fraction of the full system. GPU usage and computation time include pre-processing overhead for graph construction at each rollout step. Top-left plot reports peak GPU usage across graph sizes (R=radius, S=sample ratio); top-right shows net computation time for a rollout as a function of sparsity at fixed radius. Bottom-left plot reports rollout MSE, showing that performance remains stable ($\sim$1e-4) for sparsity levels 0.125×, 0.062×, and 0.031×. Below this, performance degrades due to oversparsification. Bottom-right plot shows that for high sparsification (0.007×), increasing the graph radius beyond 0.15 does not improve performance.

**Failure modes of ROM**  Performance degradation in ultra-sparse regimes is observed in our approach: when the sampled particle count drops such that the active voxel count approaches 0, the estimator becomes ill-conditioned, forcing the network to rely entirely on the prior, which lacks local physical guidance. We provide more discussions and an analysis of limitations in section M.

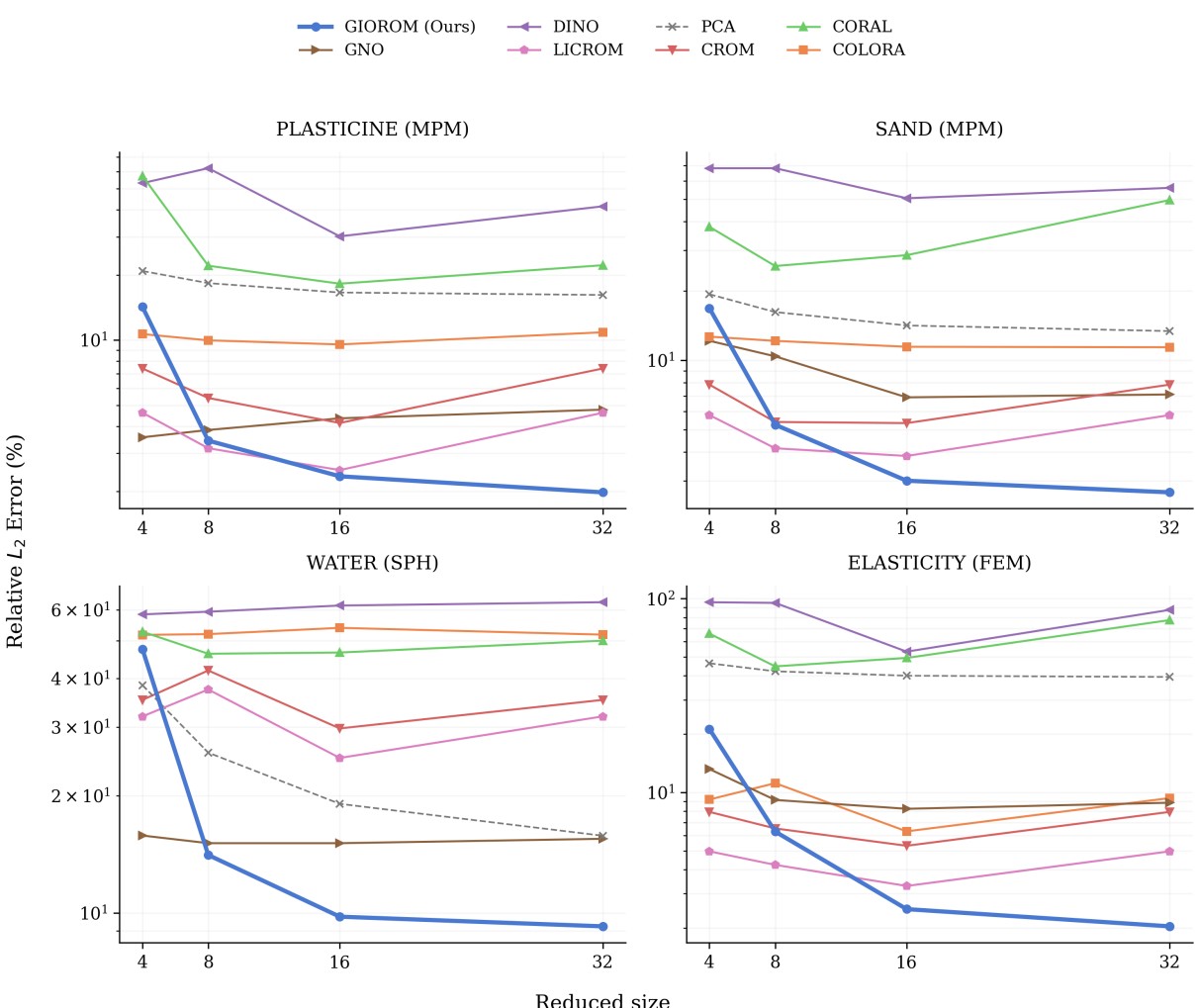

Figure 6: **Impact of ROM space on different physical regimes** The above plots showcase the effect of reduced space sparsity on 4 different regimes, ranging from highly dynamic fluids (`Water-3D`) to regimes with minimal deformation (`Plasticine-3D`) and elastic behaviors (`Elasticity-3d`). The sharp degradation in GIOROM performance observed for fewer than 8 is attributed to extreme sparsity in the Lagrangian sampling, which results in empty support neighborhoods for the kernel integration.

Table 4: Contrasting the change in computation time with the increase in connectivity radius for a graph with 7056 points. The times shown represent the overall time needed to infer all 200 time steps. We compare our time-stepper with other neural network based physics solvers.

| MODEL | TIME STEPS | NUMBER OF SPATIAL POINTS | CONNECTIVITY RADIUS | | | | | | |
|---|---|---|---|---|---|---|---|---|---|
| | | | 0.040 | 0.050 | 0.060 | 0.070 | 0.080 | 0.090 | 0.100 |
| OURS | 200 | 7056 points | **20.1s** | **34.3s** | **47.6s** | **65.8s** | **89.7s** | **104.1s** | **109.3s** |
| GNS | 200 | 7056 points | 43.5s | 73.5s | 111.6s | 162s | 226.2s | 305.9s | 386.0s |

**Limitations and Generalization** We show in our work that we can scale to millions of points and model 4 physical regimes. We additionally provide results for 3 generalization cases in fig. 21 - long horizon dynamic trajectories, multi-object elastic collisions and phase changes (melting).

GIOROM, like other learning-based ROMs, exhibits reduced performance under extreme out-of-distribution conditions (Li et al., 2020b; Chen et al., 2023) and extreme sparsifications (section M.2). Furthermore,

Table 5: Contrasting the change in computation time with an increase in graph size at a fixed radius of 0.060. The times shown represent the overall time needed to infer 200 time steps. We compare our time-stepper against other neural network based physics solvers

| MODEL | PARAMETERS | CONNECTIVITY | MATERIAL | TIME STEPS | GRAPH SIZE | | | |
| | | | | | 1776 POINTS | 4143 POINTS | 5608 POINTS | 7056 POINTS |
|---|---|---|---|---|---|---|---|---|
| OURS | 4,312,247 | 0.060 | Plasticine | 200 | **3.9s** | **14.5s** | **27.3s** | **47.6s** |
| GNS | 1,592,987 | 0.060 | Plasticine | 200 | 7.8 s | 38.3s | 68.7s | 111.6s |

while currently designed for continuous systems, future work may extend GIOROM to explicitly address discontinuities (Belhe et al., 2023; Goswami et al., 2022) and unbounded flows. While we currently only leverage Euler integration schemes, studying the impacts of higher-order methods, such as Runge-Kutta 4th order on accuracy or stability of ROM is an exciting future direction. We also restrict our problem setups to consistent time-discretizations. Understanding the behavior of neural time-steppers on varying time discretization is another consideration for future work.

**Conclusion** We show that our reduced-order modeling framework for learning Lagrangian dynamics on sparse inputs achieves computational improvements over existing neural solvers (table 5) while preserving high simulation fidelity across diverse physical systems, outperforming traditional ROM approaches, particularly, on dynamic fluid simulations. The end-to-end data-driven approach can open doors to incorporating model-order reduction to simulations generated by generative models, with no PDE priors.

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

# A    Appendix

# B    Additional related works

**Time series dynamical systems**   Simulating temporal dynamics in an auto-regressive manner is a particularly challenging task due to error accumulations during long rollout Wikner et al. (2024); List et al. (2024). There have been many works that learn temporal PDEs and CFD, including Majid & Tudisco (2024); Liu et al. (2024b); Sarkar et al. (2024); Wu et al. (2024b); Jeon et al. (2024); Jiang et al. (2024); Ma et al. (2024); Janny et al. (2024). Some works have proposed neural network-based approaches to model 3D Lagrangian dynamics, such as Ummenhofer et al. (2020), who propose a convolutional neural network-based approach to model the behavior of Newtonian fluids in 3D systems. Sanchez-Gonzalez et al. (2020) propose a more general graph-based framework, but the network suffers from high computation time on very dense graphs.

# C    Definitions

**Full-order model**   A full-order model evolves the solution over all degrees of freedom present in the spatial discretization scheme (Hughes, 2012) (e.g., $P \cdot d$ in $\mathbb{R}^{P \cdot d}$ or $r \cdot d$ in $\mathbb{R}^{r \cdot d}$). These models do not perform any dimensionality reduction. They are therefore prohibitively slow when $P$ is large. We denote the $P$-point discretization of the spatial field $\boldsymbol{X}$ by $\{x^j\}_{j=1}^{P}$, where $x^j$ represents a particle in $\mathbb{R}^d$ with negligible mass. Similarly, a sample set $\{x^i\}_{i=1}^{r} \subset \{x^j\}_{j=1}^{P}$ forms an $(r \cdot d)$ vector.

**Reduced-order model**   A reduced-order model evolves the solution in reduced latent space $\mathbb{R}^Q$, $Q \ll P \cdot d$. Reduced-order techniques such as Chen et al. (2023) leverage neural fields and projection-based ROM to infer the continuous spatial function at arbitrary spatial locations from the low-dimensional latents. In a similar vein, for an evaluation point $x^i$ and its local spatial neighborhood $\tilde{x} = \mathcal{N}(x^i)$, we define our parameterization $\kappa(x^i, \phi_\Theta) \approx \boldsymbol{f}(x^i, t)$ as $\kappa(x^i, \phi_{\Theta, \tilde{x}}) \approx \boldsymbol{f}(x^i, t)$, for $x^i \in \mathbb{R}^d$, $\tilde{x} = \mathcal{N}(x^i)$, $\phi \in \mathbb{R}^{r \cdot d}$. We note here that $\kappa$ parameterizes $\mathcal{M}(\boldsymbol{f})$ and $\phi_{\Theta, \tilde{x}} := \{\hat{\boldsymbol{f}}^j\}_{j \in \mathcal{N}(x^i)}$. We also note that $\hat{\boldsymbol{f}}$ has a temporal dependency, i.e. $\hat{\boldsymbol{f}}_t$ represents the evaluation at $t$. However, we omit the time subscript for brevity.

Traditional ROM approaches define the latents to be a nonlinear low-dimensional manifold Chen et al. (2021). Such a low-dimension parameterization can be defined with a projection mapping, $g : \mathbb{R}^Q \to \mathbb{R}^{P \cdot d}$, where $Q \ll P \cdot d$. This defines a map to a discrete field for every low dimensional latent vector $\hat{\mathbf{x}}(t) \in \mathbb{R}^Q$ such that $g_P(\hat{\mathbf{x}}) \mapsto (\boldsymbol{f}_t^1, ..., \boldsymbol{f}_t^P)$ Chen et al. (2023); Chang et al. (2023). For point-wise evaluations, $g(x^i, \hat{\mathbf{x}}_t) \approx \boldsymbol{f}(x^i, t) = \boldsymbol{f}_t^i$, with $\hat{\mathbf{x}}$ serving as a low-dimensional latent representation of the continuous field $\boldsymbol{f}$.

Discretization invariant ROM approaches Chen et al. (2023); Chang et al. (2023) construct the latent state $\hat{\mathbf{x}}_t$ from a known set of point-wise solution values $\phi = (\boldsymbol{f}_t^1, \dots, \boldsymbol{f}_t^P)$ through a projection map $\pi : \mathbb{R}^{P \cdot d} \to \mathbb{R}^Q$, often parameterized using PointNet-style models which enable continuous evaluation. However, temporal evolution, $\hat{\mathbf{x}}_t \mapsto \hat{\mathbf{x}}_{t+1}$, is implemented via explicit numerical time-stepping schemes with time gradients provided by the exact PDE. As a result, these methods depend on explicit access to both the PDE solution and the associated solver framework.

**Time integration**  To compute temporal dynamics on these $r$-point spatial samples, we seek to evolve $\{\boldsymbol{f}_t^j\}_{j=1}^r \mapsto \{\boldsymbol{f}_{t+1}^j\}_{j=1}^r$. In the discrete setting, we leverage an explicit Euler time integrator (Ascher & Petzold, 1998) with step-size $\Delta t$,

$$\boldsymbol{f}_{t+1}^j = \boldsymbol{f}_t^j + \Delta t \; \dot{\boldsymbol{f}}_t^j \tag{6}$$

$$\dot{\boldsymbol{f}}_{t+1}^j = \dot{\boldsymbol{f}}_t^j + \Delta t \; \ddot{\boldsymbol{f}}_t^j \tag{7}$$

If $\boldsymbol{f}^j$ represents the position field, then, the one and only unknown in the equation above is the acceleration $\mathbf{A}_t^j = \ddot{\boldsymbol{f}}_t^j$, which is necessary for computing the velocity $\mathbf{V}_{t+1}^j = \dot{\boldsymbol{f}}_{t+1}^j$. We learn the acceleration field $\mathbf{A}_t^j := \ddot{\boldsymbol{f}}_t^j$ using the parameterization $\phi_\Theta$, which is then used in the explicit Euler update.

## D  Background

### D.1  Operator learning

Here, we summarize the important ingredients of neural operators. For more details, please refer to Li et al. (2020a). Operator learning is a machine learning paradigm where a neural network is trained to map between infinite-dimensional function spaces. Let $\mathcal{G} : \mathcal{V} \to \mathcal{A}$ be a nonlinear map between the two function spaces $\mathcal{V}$ and $\mathcal{A}$. A neural operator is an operator parameterized by a neural network given by $\mathcal{G}_\theta \colon \mathcal{V} \to \mathcal{A}, \quad \theta \in \mathbb{R}^z$, that approximates this function mapping in the finite-dimensional space. The learning problem can be formulated as $\min_{\theta \in \mathbb{R}^z} \mathbb{E}_{v \sim D} \left[ \|\mathcal{G}_\theta(v) - \mathcal{G}(v)\|_\mathcal{V}^2 \right]$, where $\| \cdot \|_\mathcal{V}$ is a norm on $\mathcal{V}$ and $D$ is a probability distribution on $\mathcal{V}$. In practice, the above optimization is posed as an empirical risk-minimization problem, defined as $\min_{\theta \in \mathbb{R}^P} \frac{1}{N} \sum_{i=1}^N \|\mathcal{G}_\theta(v^{(i)}) - a^{(i)}\|_\mathcal{V}^2$.

Our **data-driven discretization invariant** reduced-order modeling framework for Lagrangian systems is constructed by parameterizing the PDE solution operator $\mathcal{J}$ with a data-driven neural parameterization $\phi_\Theta : a \to \hat{\boldsymbol{f}}$, trained from a small set of point-wise samples $\{\boldsymbol{a}_t^i, \boldsymbol{f}_t^i\}_{i=1}^r$ such that $\hat{\boldsymbol{f}}_t \approx \boldsymbol{f}_t$ and $\phi_\Theta \in \mathbb{R}^{r \cdot d}$, where $r \cdot d \ll P \cdot d$. We interpret $\phi_\Theta$ to be a surrogate for the solution operator, with $\{x^i\}_{i=1}^r$ representing a form of reduction of $(P \cdot d)$-point samples in $\boldsymbol{X}$. The architecture for $\phi_\Theta$ leverages graph interaction network Battaglia et al. (2016); Sanchez-Gonzalez et al. (2020) to capture particle interaction dynamics. Figure 2 illustrates this distinction by comparing particle trajectories in elastic deformation systems modeled using our approach with discretization invariant CROM Chen et al. (2023).

### D.2  Kernel methods and manifold learning

Kernel methods have been applied in manifold learning, where they serve as tools for constructing operators that act on functions defined over sampled data. Many such methods—including diffusion maps and Laplacian eigenmaps—can be interpreted as variants of kernel PCA Izenman (2012).

Belkin and Niyogi Belkin & Niyogi (2003) and Coifman et al. Coifman & Lafon (2006) studied *radially symmetric* kernels of the form $k(x, y) = h\left(\frac{\|x - y\|^2}{\varepsilon}\right)$ where $h : \mathbb{R}_+ \to \mathbb{R}$ is a decreasing function, typically Gaussian. These kernels are *local*. As such, they are isotropic and spatially invariant Berry & Sauer (2016).

Neural operator literature defines the kernel based **kernel integral transform**:

$$(\mathcal{K}f)(x) = \int_\Omega k(x, y) f(y) \, dy,$$

which is central to both manifold learning and neural operator frameworks. It maps input functions $f : \Omega \to \mathbb{R}^d$ to output functions via integration against $k$, and serves as the foundation for approximating function-to-function mappings.

In the context of manifold learning, local radially symmetric kernels induce a geometry on the embedded manifold. As the sample density increases, the kernel implicitly defines a metric structure through its interactions over neighborhoods on the manifold Berry & Sauer (2016). Consequently, such kernels not only enable dimensionality reduction but also act as geometric priors over data manifolds.

Finally, this kernel framework extends naturally to operator learning, where local solution maps, such as those arising from hyperbolic PDEs, can be effectively approximated using locally supported kernels Liu-Schiaffini et al. (2024).

## E  Consistency & stability

To analyze the theoretical properties of the proposed GIOROM, we study the error decomposition of the discretized integral operator. The continuous operator is defined as $\mathcal{H}[f](\mathbf{x}) = \int_\Omega \kappa(\mathbf{x}, \mathbf{y}) f(\mathbf{y}) \, d\mathbf{y}$. Its discrete approximation $\hat{\mathcal{H}}$ introduces error from three distinct sources: Lagrangian sampling, grid discretization, and neural approximation.

Let $u(\mathbf{x})$ denote the true drift field and $\hat{u}(\mathbf{x})$ its reconstruction produced by GIOROM.

**Monte-Carlo Consistency**  The projection of Lagrangian particles onto the grid can be interpreted as a Monte-Carlo approximation of the integral operator Caflisch (1998). Let $\{\mathbf{y}_i\}_{i=1}^N$ be i.i.d. samples drawn from a density $\rho$ on $\Omega$. Then, for any square-integrable function $f$, the Monte-Carlo estimator satisfies

$$\mathbb{E}\left[\left\| \int f(\mathbf{y}) \, d\rho(\mathbf{y}) - \frac{1}{N}\sum_{i=1}^N f(\mathbf{y}_i) \right\|\right] \leq \frac{\sqrt{\mathrm{Var}_\rho(f)}}{\sqrt{N}}, \tag{8}$$

where the expectation is taken over the sampling of $\{\mathbf{y}_i\}$. Consequently, the sampling error decays at the standard Monte-Carlo rate Robert & Casella (2004).

**Grid Convergence**  The use of trilinear splatting and interpolation corresponds to a tensor-product linear B-spline basis. Assuming $f \in C^2(\Omega)$ and a normalized partition-of-unity interpolation scheme, standard approximation theory yields the interpolation error bound

$$\|f - \Pi_{\mathrm{grid}} f\|_{L^\infty(\Omega)} \leq C \, (\Delta x)^2 \, \|\nabla^2 f\|_{L^\infty(\Omega)}, \tag{9}$$

where $\Delta x$ denotes the grid spacing and $C$ is a constant independent of $\Delta x$ Strang & Fix (1973); De Boor (1978).

**Total Error**  Combining the sampling and grid discretization errors, the estimator is consistent. As the number of particles $N \to \infty$ and the grid resolution $\Delta x \to 0$, the discrete operator converges to the continuous limit, up to the approximation error of the neural parameterization. Specifically,

$$\|\hat{u} - u\| \leq \frac{C_1}{\sqrt{N}} + C_2(\Delta x)^2 + \epsilon_\theta, \tag{10}$$

where $\epsilon_\theta$ denotes the approximation error induced by the finite expressivity and optimization of the neural networks Hornik et al. (1989).

**Sampling Condition**  To select a subset of Lagrangian markers, we employ uniform random subsampling. While less deterministic than covering-based methods, random sampling preserves the i.i.d. assumption required for the Monte Carlo consistency bound derived above. For random samples in a bounded domain $\Omega \subset \mathbb{R}^d$, the fill distance (covering radius) $h_{X,\Omega}$ converges probabilistically as $h_{X,\Omega} \sim \left(\frac{\log N}{N}\right)^{1/d}$, which ensures that the spatial coverage becomes dense as $N \to \infty$.

However, random sampling may locally violate the coverage condition in sparse regimes due to stochastic clustering. In practice, stability requires that the local particle density remains sufficient relative to the grid stencil. To quantify this, we define a heuristic condition analogous to the Particle-Per-Cell (PPC) criteria found in PIC literature Birdsall & Langdon (1991); Jiang et al. (2015): $N_{\mathrm{eff}} = \frac{N \cdot V_{\mathrm{stencil}}}{N_{\mathrm{active}}} \gtrsim 1$ This condition ensures that the average number of particles contributing to an active grid cell remains sufficient to avoid aliasing artifacts, though it does not constitute a sufficient theoretical guarantee.

### E.1 Graph interaction network

The graph interaction network proposed in Battaglia et al. (2016) learns a relation-centric function $f$ that encodes spatial interactions between the interacting nodes within a system as a function of their interaction attributes $r$. This can be represented as $e_{t+1} = f_R(x_{1,t}, x_{2,t}, r)$. A node-centered function predicts the temporal dynamics of the node as a function of the spatial interactions as follows $x_{1,t+1} = f_o(e_{t+1}, x_{1,t})$. In a system of $m$ nodes, the spatial interactions are represented as a graph, where the neighborhood is defined by a ball of radius r. This graph is represented as $G(O, R)$, where $O$ is the collection of objects and $R$ is the relationships between them. The interaction between them is defined as $\mathcal{I}(G) = f_o(a(G, X, f_R(\langle x_i, x_j, r_{ij} \rangle)))$, Where $a$ is an aggregation function that combines all the interactions, $X$ is the set of external effects, not part of the system, such as gravitational acceleration, etc.

## F Algorithms

The following section presents key algorithms that are helpful for implementing the concepts presented in the paper

---

**Algorithm 1** GNN Time-Stepper Inference

---

**Input:**
$G = (V, E)$: input graph with nodes $V$ and edges $E$
$x \in \mathbb{N}^{|V| \times 1}$: categorical node types
$p \in \mathbb{R}^{|V| \times d}$: recent positions and velocities
$e \in \mathbb{R}^{|E| \times d_e}$: edge attributes (displacement, distance)
$\texttt{edge\_index} \in \mathbb{N}^{2 \times |E|}$: sender and receiver indices
$\texttt{recent\_pos} \in \mathbb{R}^{|V| \times d}$: current position of each node
**Output:**
$\hat{a} \in \mathbb{R}^{|V| \times r}$: predicted acceleration (or equivalent output)

  1: **Step 1: Node and Edge Feature Initialization**
  2: $h_v \leftarrow \texttt{concat}(\texttt{EmbedType}(x), p)$           ▷ Embed categorical type and concatenate with pos/velocity
  3: $h_v \leftarrow \texttt{NodeInputMLP}(h_v)$
  4: $h_e \leftarrow \texttt{EdgeInputMLP}(e)$
  5: **Step 2: Message Passing (Encoder)**
  6: **for** $i = 1$ **to** $\texttt{n\_mp\_layers}$ **do**
  7:     $h_v, h_e \leftarrow \texttt{GNNLayer}_i(h_v, \texttt{edge\_index}, h_e, \texttt{node\_dist})$
  8: **end for**
  9: **Step 3: Global Transformation via GNOT Layer**
10: $h_v \leftarrow \texttt{GNOTLayer}(h_v, \texttt{recent\_pos})$
11: **Step 4: Message Passing (Decoder)**
12: **for** $i = 1$ **to** $\texttt{n\_mp\_layers}$ **do**
13:     $h_v, h_e \leftarrow \texttt{GNNLayerOut}_i(h_v, \texttt{edge\_index}, h_e, \texttt{node\_dist})$
14: **end for**
15: **Step 5: Node-wise Output Projection**
16: $\hat{a} \leftarrow \texttt{NodeOutputMLP}(h_v)$
17: **return** $\hat{a}$

---

## G Architecture outline

The schematic in fig. 7 illustrates the full sequence of operations performed during a single time-step of the proposed method. It delineates the interaction between the time-stepping mechanism, the kernel-based reduced-order model (kernel-integral ROM), the Euler integration scheme, and the spatial sampling strategy used to update the system state.

---

**Algorithm 2** Learnable Kernel-Integral ROM (GIOROM)

---

**Require:** Source state $\{\mathbf{x}_s, \mathbf{u}_s\}$, Query points $\mathbf{x}_q$, Grid Res $R$, Noise $\sigma$, Paths $K$
**Ensure:** Predicted drift field $\hat{\mathbf{u}}_q$
 1: **1. Feature Encoding**
 2: $\mathbf{v}_s \leftarrow E_\theta(\mathbf{u}_s)$              ▷ Lift drift to latent features (MLP)
 3: **2. Grid Splatting (Monte-Carlo Integration)**
 4: Initialize grid $\mathbf{G} \in \mathbb{R}^{R \times R \times R \times C}$ with zeros
 5: **for** each particle $p$ in source set **do**
 6:    $\mathbf{G} \leftarrow \mathbf{G} + \mathbf{v}_p \cdot \Lambda(\mathbf{x}_p)$         ▷ Accumulate features via trilinear weights
 7: **end for**
 8: **3. Stochastic Sampling (Feynman-Kac)**
 9: **for** each query point $j$ in $\mathbf{x}_q$ **do**
10:    Sample noise $\xi_1, \ldots, \xi_K \sim \mathcal{N}(0, \sigma^2 \mathbf{I})$
11:    $\mathbf{h}_j \leftarrow \frac{1}{K} \sum_{k=1}^{K} \text{Interp}(\mathbf{G}, \mathbf{x}_j + \xi_k)$         ▷ Convolve & Sample
12:    **4. Residual Decoding**
13:    $\hat{\mathbf{u}}_j \leftarrow D_\theta(\mathbf{h}_j) + \text{Interp}(\mathbf{u}_s, \mathbf{x}_j)$        ▷ Decode & Add Residual
14: **end for**
    return $\hat{\mathbf{u}}_q$

---

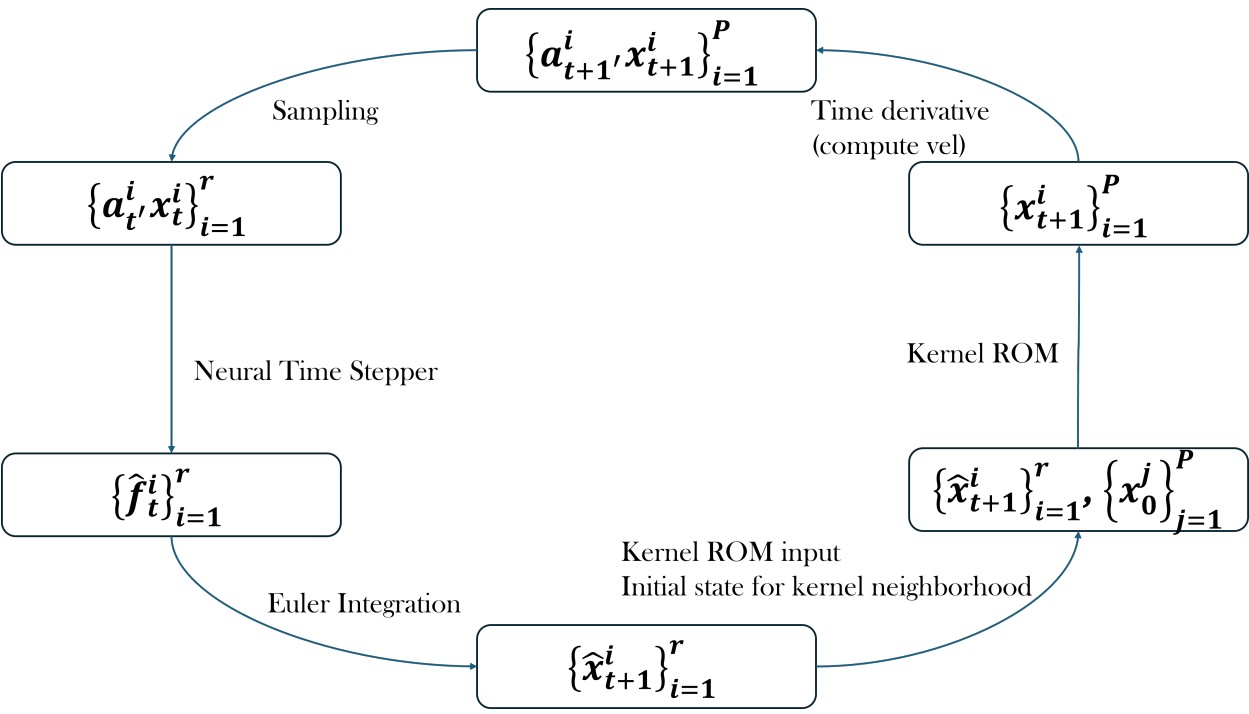

Figure 7: **Single-step flow diagram:** Overview of the forward pass used to infer the system state at the next time-step via kernel-integral ROM, time-stepping, Euler integration, and spatial sampling. As $\phi_\Theta$ takes as input a window of past-states, we require the same set of samples for a trajectory evolution

## H Experimental setup details

### H.1 PDE operator setup and hyperparameters

**Data representation** To train $\phi_\Theta$, we create a window of $w$ point cloud velocity sequences as the input defined on the nodes, with the pointwise acceleration as the output. We define $\{\mathbf{x}_t^i\} \in \mathbb{R}^{r \cdot d}$ to be the pointwise positions of $r$ particles within a $d$-dimensional system at time $t$. A sequence of $T$ time steps is denoted as

$\mathbf{x}_{0:T} = (\mathbf{x}_0, \ldots, \mathbf{x}_T)$. In particular, $\{\mathbf{x}_t^0, \ldots, \mathbf{x}_t^r\}$ are the individual particles within the system at time $t$. We define velocity at time $n$ as $\mathbf{v}_t$ as $\mathbf{x}_t - \mathbf{x}_{t-1}$. Similarly, acceleration at time $n$ is defined as $\mathbf{a}_t = \mathbf{v}_t - \mathbf{v}_{t-1}$. In all these cases, $\Delta t$ is set to one for simplicity. To train the model, we follow the same procedure as (Sanchez-Gonzalez et al., 2020). We additionally encode the particle types (water, sand, plasticine, etc.) as embeddings, which is useful for multimaterial simulations. The graph is constructed as a `radius_graph`, defined within `pytorch geometric` library.

**Boundary representation** To enforce the boundaries of the system, the node feature includes the past $w$ velocity fields as well as the distance of the most recent position field to the upper ($b_u$) and lower ($b_l$) boundaries of the computational domain, given by $\mathcal{D} = [(x - b_l)/\rho, (b_u - x)/\rho], \forall x$, where $\rho$ is the radius of the graph.

## H.2 Hyperparameters

The models were implemented using `Pytorch` library and trained on `CUDA`. The graphs were built using `Pytorch Geometric` module. All models were trained on `NVIDIA RTX 3060` GPUs.

$\phi_\Theta$ **time-stepper** The graph is constructed using `radius_graph` defined in `Pytorch Geometric`. The node features and the edge features, which include the distance from the boundary points, are encoded into latent vectors of size 128 using 2 MLPs. The encoder uses two layers of interaction network. The latents are then processed by two layers of Neural Operator Transformer. The decoder layers are symmetric to the encoder layers. The NOT block uses 4 attention heads, branch and trunk sizes of 32, and an output dim of 128.

**Learnable Kernel-Integral ROM** We implement the kernel integration using `Jax` with a differentiable, grid-based Monte-Carlo approximation. Unlike standard grid methods, we interpret the Lagrangian particles not just as mass points, but as Monte-Carlo samples of an underlying continuous latent field. The architecture approximates the integral transform $\mathcal{H}[\mathbf{f}](\mathbf{x}) = \int \kappa(\mathbf{x}, \mathbf{y})\mathbf{f}(\mathbf{y})d\mathbf{y}$ through a learnable three-stage pipeline parameterized by $\theta = \{\mathbf{W}_{\text{enc}}, \mathbf{W}_{\text{dec}}\}$:

**Learnable Feature Encoding:** First, we lift the raw Lagrangian drift $\mathbf{u}_p = \mathbf{x}_p^{(t)} - \mathbf{x}_p^{(0)}$ into a high-dimensional latent feature space. This mapping is parameterized by a shallow MLP encoder $E_\theta$ comprising two hidden layers of width 64 with GELU activations:

$$\mathbf{v}_p = E_\theta(\mathbf{u}_p) = \text{GELU}(\mathbf{W}_{\text{enc}}\mathbf{u}_p + \mathbf{b}_{\text{enc}}), \tag{11}$$

where $\mathbf{v}_p \in \mathbb{R}^{d_f}$ represents the learned feature weight for particle $p$. We set the latent feature dimension $d_f = 4$ (3 learned features + 1 density channel) to balance expressivity with memory efficiency.

**Grid Splatting & Monte-Carlo Integration:** We approximate the continuous field $\Psi(\mathbf{x})$ on a regular Euclidean grid $G$ of resolution $64 \times 64 \times 64$ covering the normalized domain $\Omega = [0, 1]^3$. This step acts as a discretized Monte-Carlo integration of the particle features onto the grid manifold:

$$\mathbf{G_i} = \sum_{p \in \mathcal{N}(\mathbf{i})} \mathbf{v}_p \cdot \Lambda(\mathbf{x}_p - \mathbf{x_i}), \tag{12}$$

where $\mathbf{G_i}$ is the feature vector at grid vertex $\mathbf{i}$, and $\Lambda(\cdot)$ is the trilinear basis function. To enforce smoothness and emulate a continuous Gaussian kernel operator, we apply a stochastic convolution step during sampling. We define the smoothed field value at query point $\mathbf{x}_q$ as the expectation over a noise distribution $\xi \sim \mathcal{N}(0, \sigma^2 I)$:

$$\hat{\mathbf{v}}(\mathbf{x}_q) = \mathbb{E}_\xi \left[ \text{Interp}(\mathbf{G}, \mathbf{x}_q + \xi) \right]. \tag{13}$$

In practice, equation 13 is approximated via Monte-Carlo sampling with $K = 8$ paths per query, effectively convolving the grid features with a Gaussian kernel of width $\sigma = 0.42$ (in grid units).

**Residual Decoding:** Finally, the smoothed grid features are mapped back to the physical drift space via a decoder MLP $D_\theta$ (width 64, GELU). To preserve high-frequency details lost during the smoothing step, we employ a residual connection:

$$\hat{\mathbf{u}}_q = D_\theta(\hat{\mathbf{v}}(\mathbf{x}_q)) + \mathbf{u}_{\text{interp}}, \tag{14}$$

where $\mathbf{u}_{\text{interp}}$ is the direct trilinear interpolation of the drift field. The entire pipeline is trained end-to-end with Adam optimization (LR=$10^{-3}$) by minimizing the $L^2$ reconstruction loss $\mathcal{L} = \sum_q \|\hat{\mathbf{u}}_q - \mathbf{u}_q^{\text{GT}}\|^2$.

**Optimizers**   Optimization is done with Adamax optimizer, with an initial learning rate of 1e-4, weight decay of 1e-6 and a batch size of 4. The learning rate was decayed exponentially from $10^{-4}$ to $10^{-6}$ using a scheduler, with a gamma of $0.1^{1/5e6}$

## I   Additional dataset details

We model the following classes of materials - elastic, plasticine, granular, Newtonian fluids, non-Newtonian fluids, and multi-material simulations. Notably, this framework is compatible with data generated using different solvers such as Finite Element Method (FEM) Hughes (2012), Material Point Method (MPM) Jiang et al. (2016), or Smooth Particle Hydrodynamics (SPH) Monaghan (1992).

**Plasticine (von Mises Yield)**   Using the `NCLAW` simulator, we generated 100 trajectories of 400 time steps ($dt = 5e - 4$) with random initial velocities and 4 different geometries - Stanford bunny, Stanford armadillo, blub (goldfish), and spot (cow). The trajectories are modeled using Saint Venant-Kirchoff elastic model, given by

$$\mathbf{P} = \mathbb{U}(2\mu\epsilon + \lambda tr(\epsilon))\mathbb{U}^T \tag{15}$$

where $\lambda$ and $\mu$ are Lamé constants, $\mathbf{P}$ is the second Piola-Kirchoff stress and $\epsilon$ is the strain. $\mathbb{U}$ is obtained by applying SVD to the deformation gradient $\mathbf{F} = \mathbb{U}\mathbf{\Sigma}\mathbf{V^T}$. The von Mises yield condition is denoted by

$$\delta\gamma = \|\hat{\epsilon}\| - \frac{\tau_Y}{2\mu} \tag{16}$$

where $\epsilon$ is the normalized Henky strain, $\tau_Y$ is the yield stress.

**Granular material (Drucker Prager sand flows)**   We trained the model on 2 datasets to simulate granular media. We generated 100 trajectories at 300 time steps, using `NCLAW` simulator and on the 2D Sand dataset released by Pfaff et al. (2020). The Drucker-Prager elastoplasticity is modeled by the same Saint Venant–Kirchhoff elastic model, given by Equation 15. Additionally, the Drucker-Prager yield condition is applied such that

$$tr(\epsilon) > 0 \quad or \quad \delta\gamma = \|\hat{\epsilon}\| + \alpha\frac{(3\lambda + 2\mu)tr(\epsilon)}{2\mu} > 0 \tag{17}$$

where, $\alpha = \sqrt{2/3}\frac{2sin\theta}{3-sin\theta}$ and $\theta$ is the frictional angle of the granular media.

**Elasticity**   To simulate elasticity, we generated simulations using meshes from Thingi10k dataset Zhou & Jacobson (2016). We generated 24 trajectories, with 200 time steps, for 6 geometries to train the model. The elasticity is modeled using stable neo-Hookean model, as proposed in Smith et al. (2018). The energy is denoted by

$$\Psi = \frac{\mu}{2}(I_C - 3) + \frac{\lambda}{2}(J - \alpha)^2 - \frac{\mu}{2}log(I_C + 1) \tag{18}$$

where $I_C$ refers to the first right Cauchy-Green invariant and $J$ is the relative volume change. $\mu$ and $\lambda$ are Lamé constants. The corresponding Piola-Kirchoff stress is given by

$$\mathbf{P} = \mu\Big(1 - \frac{1}{I_C + 1}\Big)\mathbf{F} + \lambda(J - \alpha)\frac{\partial J}{\partial \mathbf{F}} \tag{19}$$

where $\mathbf{F}$ is the deformation gradient.

**Newtonian fluids**   For Newtonian fluids, In the 2D setting, we use `WaterDrop` dataset created by Pfaff et al. (2020), which is generated using the material point method (MPM). For the 3D setting, we generated 100 trajectories with random initial velocity, each spanning 1000 time steps at a dt of $5e-3$. This dataset was prepared using the `NCLAW` framework. These are modeled as weakly compressible fluids, using fixed corotated elastic model with $\mu = 0$. The Piola-Kirchoff Stress is given by

$$\mathbf{P} = \lambda J(J - 1)\mathbf{F}^{-T} \tag{20}$$

**Non-Newtonian fluids** To train the model on non-Newtonian fluids, we used the `Goop` and `Goop-3D` datasets.

**Multimaterial** We simulated multi-material trajectories in 2D using the dataset published by Pfaff et al. (2020).

## J Training setup

**Time-stepper** We follow the exact training procedure outlined in Sanchez-Gonzalez et al. (2020) for $\phi_\Theta$. We use the 1-step loss function over a pair of consecutive time steps $k$ and $k+1$, imposing a strong inductive bias towards a Markovian system. Each system is trained for 5 million steps.

The model is validated by full rollouts on 10 held-out validation sets per material simulation, with performance measured by the MSE between predicted particle positions and ground-truth particle positions.

**kernel-integral ROM** To train the kernel-integral ROM, we define the neighborhood over samples defined at $t = 0$, i.e. $\{x_0^i\}_{i=1}^P$ and $\{x_0^j\}_{j=1}^r$. For all the subsequent time-steps, we leverage this neighborhood. This enables us to evolve $\{\hat{f}\}$ using the spatial information of the state at $t = 0$. The input function for the model is point-wise positions at a given time-step $t$ for non-fluid systems and deformation $x_t - x_0$ for fluid systems. We generate these using $\phi_\Theta$. Each system is trained for 30,000 steps and evaluated on unseen discretizations and trajectories.

## K Additional results

**2D simulations** Figure 8 presents qualitative comparisons between ground truth and $(P \cdot d)$-point predictions produced by GIOROM on various 2D simulations. The subfigures depict a range of dynamic behaviors: (a) granular flow, (b) soft body motion under gravity, (c) external force acting on a highly elastic material, and (d) coupled interactions between granular media and Newtonian fluids.

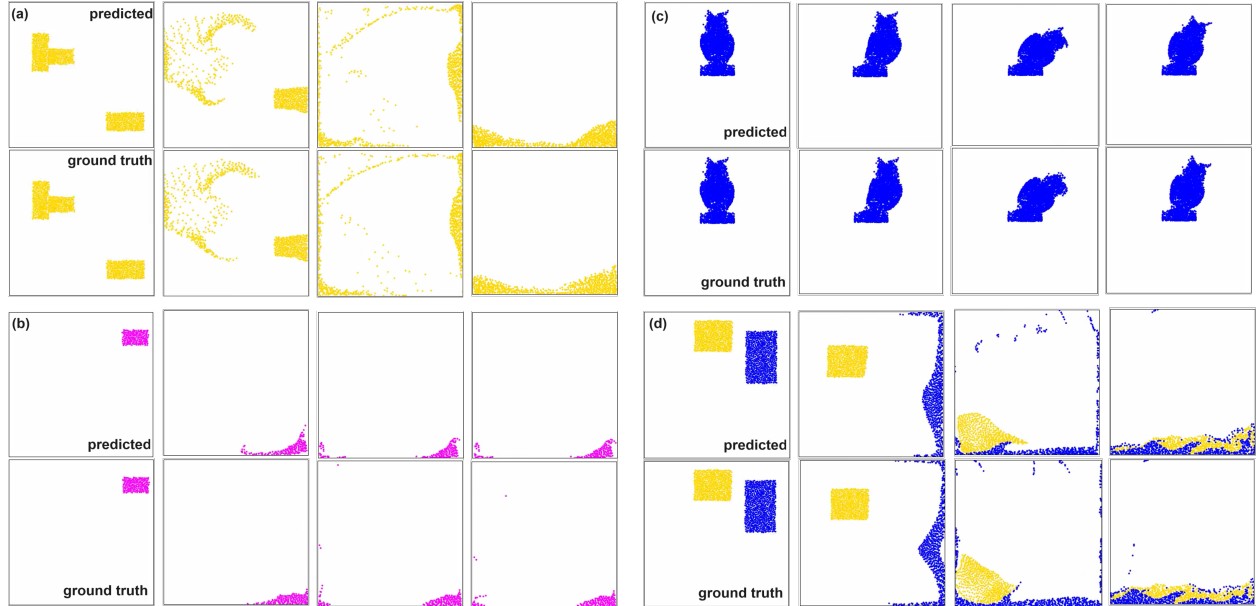

Figure 8: **2D point-cloud simulations:** $(P \cdot d)$-point inference results using GIOROM. **(a)** Granular flow; **(b)** Soft body under gravity; **(c)** Elastic response to external force; **(d)** Coupled fluid-granular interactions.

## L    Ablations

**Ablations on PDE operator**    Following Wu et al. (2024a); Alkin et al. (2024), we utilize interaction networks to model Lagrangian inputs. To justify this choice, we compare against GNS and neural operator models built for Eulerian regimes. Table 6 represents the rollout performance of different Neural Operator models on reduced sample sets. The performance is measured as the average MSE accumulated over the entire duration. We compare against **GINO (Eulerian)** Li et al. (2024), General Neural Operator Transformer **GNOT (Eulerian)** Hao et al. (2023) and Inducing Point Operator Transformer **IPOT (Eulerian)** Lee & Oh (2024). Additionally, we compare against graph neural network based model **GNS** Sanchez-Gonzalez et al. (2020). We do not consider the Lagrangian field variant of the UPT family as a baseline as it neither encodes particle interaction dynamics nor supports autoregression, making it incompatible with the datasets used in this work. All the operators struggle to generalize to fluid simulations but interaction network based models (GNS and GIOROM) exhibit the best performance.

Table 6: This table compares the rollout MSE of our PDE operator $\phi_\Theta$ against other neural PDE operators on reduced sampled sets $\mathbb{R}^{r \cdot d}$. This highlights the importance of interaction network encoders and decoders for modeling particle dynamics.

| MODEL | WATER-3D | PLASTICINE | ELASTIC | SAND-3D |
|---|---|---|---|---|
| GNS | 0.0108 | 0.0038 | 0.0003 | 0.0008 |
| GINO | 0.38 | 0.09 | 0.18 | 0.07 |
| GNOT | 0.046 | 0.0052 | 0.0028 | 0.0085 |
| IPOT | 0.15 | 0.097 | 0.084 | 0.0075 |
| $\phi_\Theta$ | 0.0106 | 0.0008 | 0.0004 | 0.0009 |

**Number of message-passing layers**    We show that the key bottleneck in terms of speed is the message-passing operation within the Interaction Network encoder and decoder.

Table 7: This table shows that the number of message-passing layers results in a negligible improvement in rollout Loss.

| NUM. MESSAGE PASSING LAYERS | CONNECTIVITY | INPUT SIZE | INFERENCE TIME/STEP | LOSS |
|---|---|---|---|---|
| 2 | 0.077 | 2247 | 3.6 | 0.0008 |
| 4 | 0.077 | 2247 | 3.8 | 0.0009 |
| 6 | 0.077 | 2247 | 4.2 | 0.0014 |
| 8 | 0.077 | 2247 | 4.3 | 0.0009 |

### L.1    Sampling strategy and rollout loss

We compared different sampling strategies against the rollout Loss (MSE). The results are presented in Table 8.

Table 8: Comparison of different sampling and graph construction strategies against rollout MSE on Water-2D dataset

| SAMPLING STRATEGY | GRAPH TYPE | ROLLOUT MSE |
|---|---|---|
| RANDOM | RADIUS | 0.0098 |
| RANDOM | DELAUNAY | 7.017 |
| FPS | RADIUS | 0.0097 |
| FPS | DELAUNAY | 8.04 |

## M   Additional discussions

### M.1   Understanding the correlation between sparsification and performance

We study the relationship between performance and sample size to understand how the structural fidelity of the interaction graph degrades the predictions under extreme sparsifications. Our analysis focuses on a small Water2D system (678 particles), where the graph structure can be explicitly visualized using `networkx`, as well as a larger Plasticine3D system. We consider different sampling levels to examine how graph sparsification impacts this relationship.

Figure 9 presents rollout MSE at two sampling ratios—35% and 11%—for the Water2D system. At 35% sampling, we observe a trend: rollout MSE consistently drops with improving the connectivity. However, with further modifications to the radius, the MSE increases. This suggests that beyond a point, the addition of edges leads to degradations. The visual rollout in fig. 10 shows that improving connectivity improves the performance upto a limit.

In contrast, at 11% sampling, the rollout MSE remain high regardless of radius, and no clear trend emerges. As shown in fig. 11, the predicted rollouts diverge significantly from the ground truth. The third column in each frame represents the reduced-space ground truth at $(r \cdot d)$ points, which itself is visually and physically distinct from the full-resolution system. This indicates that at extreme sparsity, the reduced graph fails to retain sufficient physical characteristics, rendering it unsuitable for accurate inference.

Figure 12 extends this analysis to the Plasticine3D system. At 22% sampling, we again observe that rollout MSE initially decreases, but as the radius increases further, this trend reverses. However, at 3.3% sampling, there is no correlation.

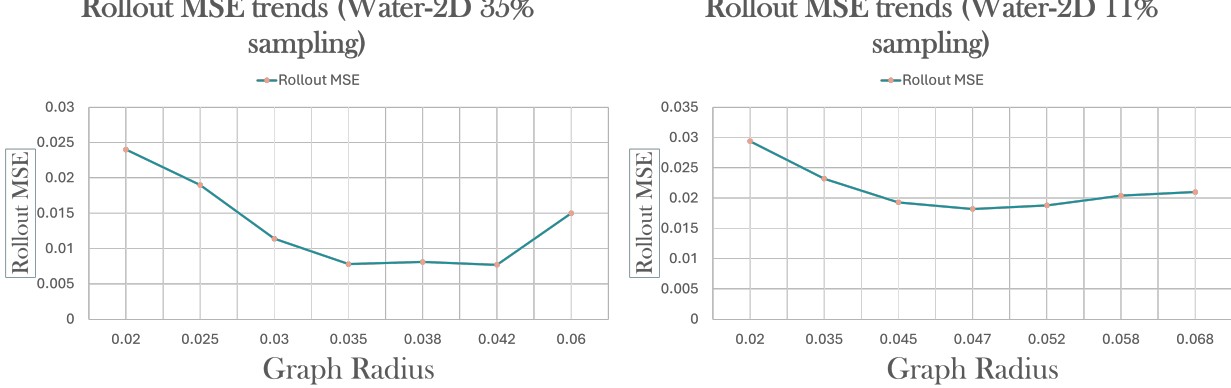

Figure 9: **Water2D: Radius vs. rollout MSE.** Left: 35% sampling. Right: At 11%, there is a weak correlation between them.

### M.2   Limitations of kernel-integral ROM to extreme sparsifications

The kernel-integral ROM model employs an implicit neural representation to approximate discrete kernel integral transforms over sparse point-wise function evaluations. However, under extreme sparsification, the quality of this representation degrades significantly.

When the number of supervised function evaluations is severely limited, increasing the neighborhood does not lead to meaningful improvements. Instead, the model interpolates around the sparse active voxels, leading to point clustering near known estimates or interpolation artifacts that resemble piecewise-linear transitions. These effects arise not due to kernel parameter choices, but due to a fundamental lack of information in the reduced space: the interpolated function no longer reflects a smooth or physically meaningful structure.

This limitation is evident in fig. 13, where we interpolate a 2-million-point Dragon mesh from three different supervision levels: 700 points (0.035%), 20K points (1%), and 60K points (3%). At 700 points, the

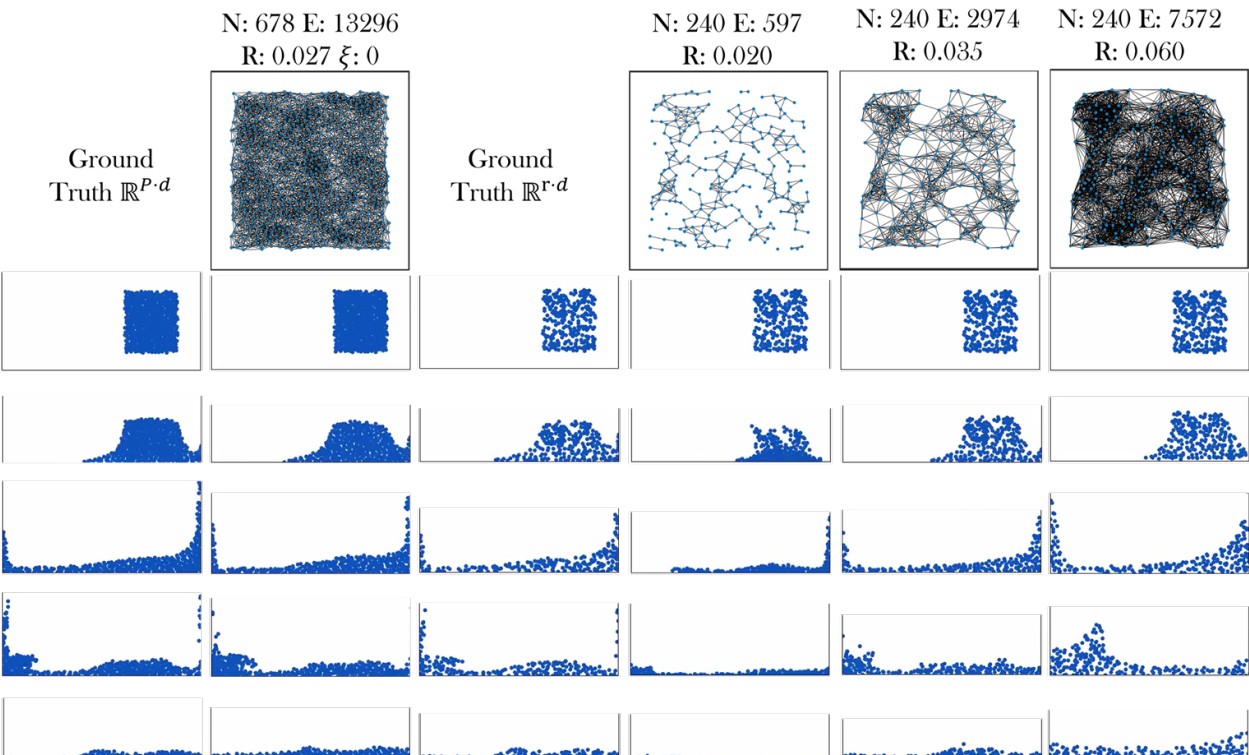

Figure 10: **Water2D at 35% sampling.** First two columns: full-space ground truth and prediction on $(P \cdot d)$ points. Third column: reduced-space ground truth at $(r \cdot d)$ points. Right columns: predictions at varying radii. Rollouts remain visually and physically consistent with the full system at R=0.035

interpolated output exhibits dense clustering and abrupt discontinuities. The gap between 0.035% and 1% remains substantial. The top-row result makes clear that in the regime of extreme sparsification, the model fails to maintain coherent spatial representations, regardless of neighborhood density or kernel formulation.

This example illustrates the lower bound on sparsity below which the learned representation fails to extrapolate. In such cases, the reduced-space support does not sufficiently reflect the full system's physical characteristics, making the reconstruction problem ill-posed.

**Sensitivity Analysis & Convergence**  To evaluate the impact of kernel parameterization, we perform an ablation study on the two primary discretization parameters: the background grid resolution $R$ and the Lagrangian sparsity factor $S$. Figure 14 summarizes the results on the Elastic (Owl) dataset.

In Figure 14(a), we observe a clear convergence behavior with respect to the grid resolution. At coarse resolutions ($R = 20$), the grid fails to resolve high-frequency deformation gradients, resulting in higher reconstruction error ($\sim 5.6\%$). As the resolution increases, the error decays rapidly, effectively plateauing around $R = 64$. This confirms that once the grid frequency exceeds the Nyquist limit of the underlying deformation field, the error becomes dominated by the neural approximation rather than spatial discretization, justifying our choice of $R = 64$ as an efficient operating point.

Figure 14(b) demonstrates the method's sensitivity to Lagrangian sampling density. Notably, the reconstruction remains robust even as the available tracking markers become extremely sparse. Increasing the sparsity factor from $12\times$ to $100\times$ (reducing the particle count by an order of magnitude) results in only a marginal increase in error (from $\sim 1.9\%$ to $\sim 3.0\%$). This stability supports our theoretical claim that the method relies on the *effective* latent support of the grid manifold rather than raw particle density. The shaded regions indicate the standard deviation across test trajectories, showing that while higher sparsity introduces slightly more variance, the method avoids catastrophic failure modes even in data-starved regimes.

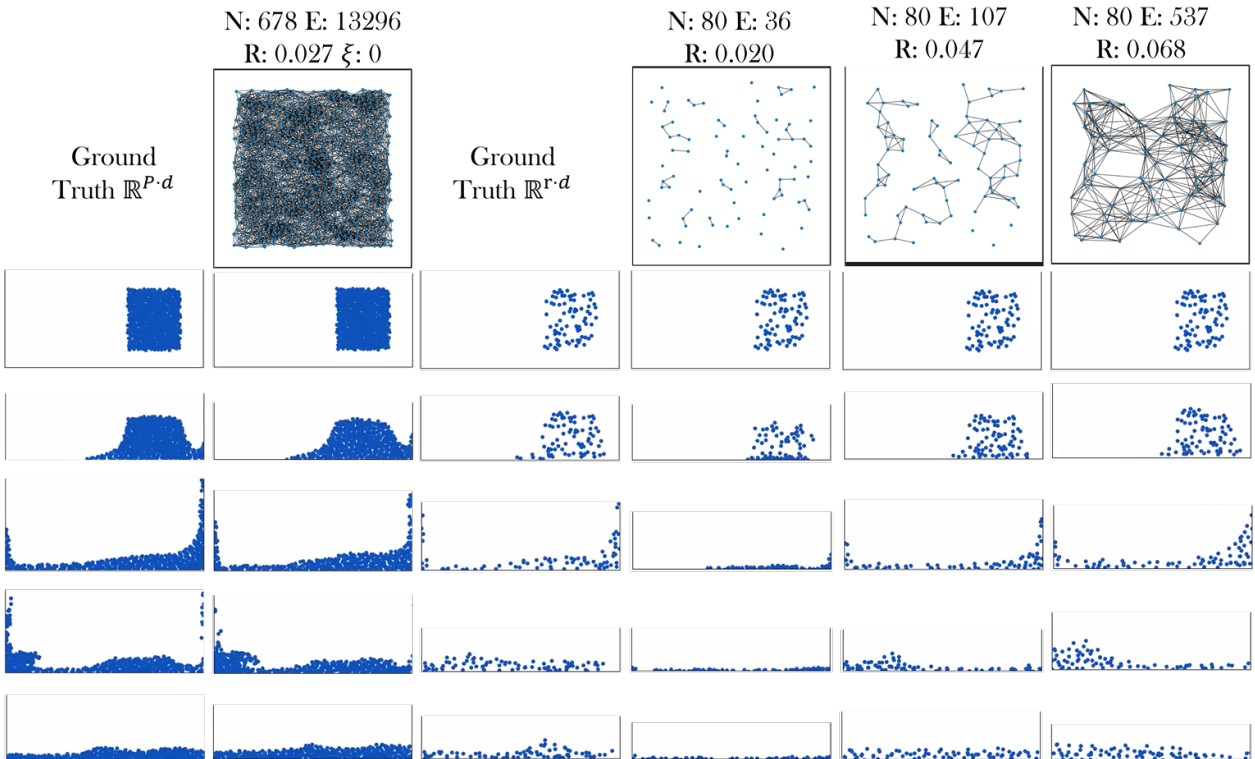

Figure 11: **Water2D at 11% sampling.** All predictions diverge from ground truth. Even at increased radii, rollout MSE remains high. Reduced-space ground truth itself is physically dissimilar to the original system, limiting learnability.

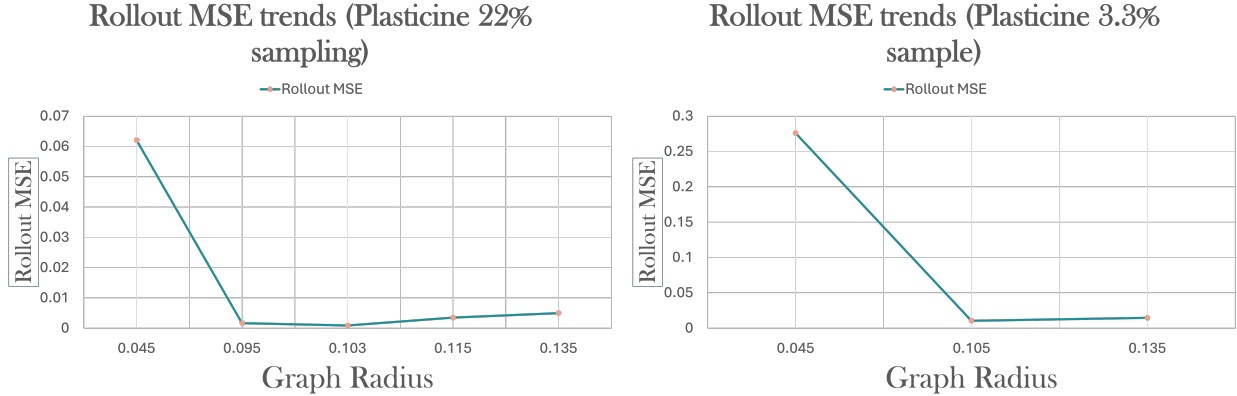

Figure 12: **Plasticine3D: rollout MSE.** Left: 22% sampling shows consistent correlation. Right: At 3.3% sampling, correlation degrades as the reduced graph fails to encode sufficient structure for accurate rollout. At 35%, rollout MSE reduces to 9.2e-4, while it plateaus at 0.01 at 3.3%

## M.3  GNNs as neural operator approximations

We now argue that Graph Neural Networks (GNNs), under a suitable construction, can be interpreted as discretizations of kernel integral operators. In particular, GNNs can be viewed as data-driven approximations to a class of neural operators. However, GNNs are senstive to input graph topologies, and cannot generalize to arbitrary graph structures defined over a discretization. This can be seen in fig. 15, where the model can

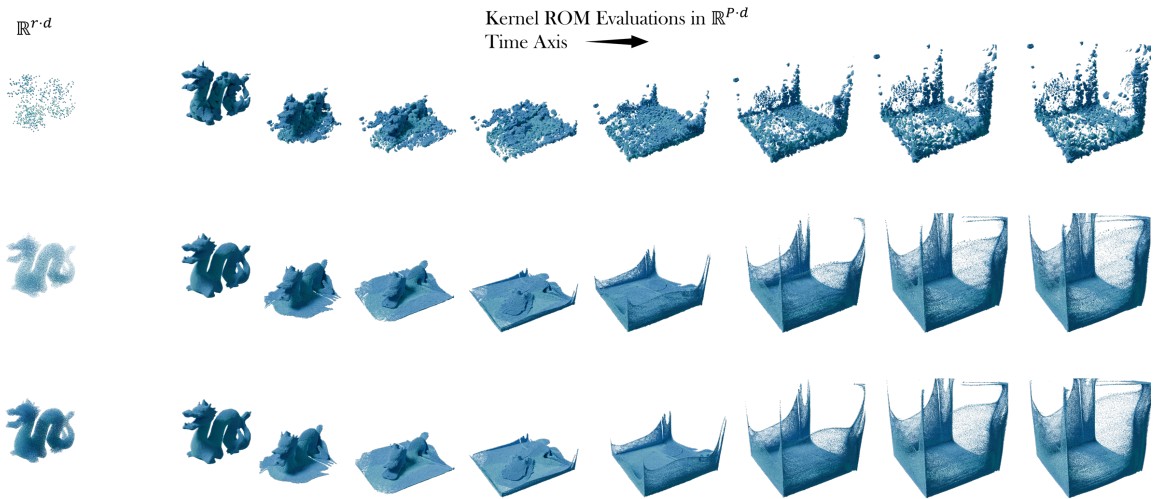

Figure 13: **Degradation under extreme sparsification.** All rows show kernel-integral ROM interpolation on a 2-million-point Dragon mesh. **Top:** 700 function evaluations (0.035%) lead to clustered and discontinuous outputs. **Middle:** 20K points (1%) **Bottom:** 60K points (3%) produces smoother reconstruction.

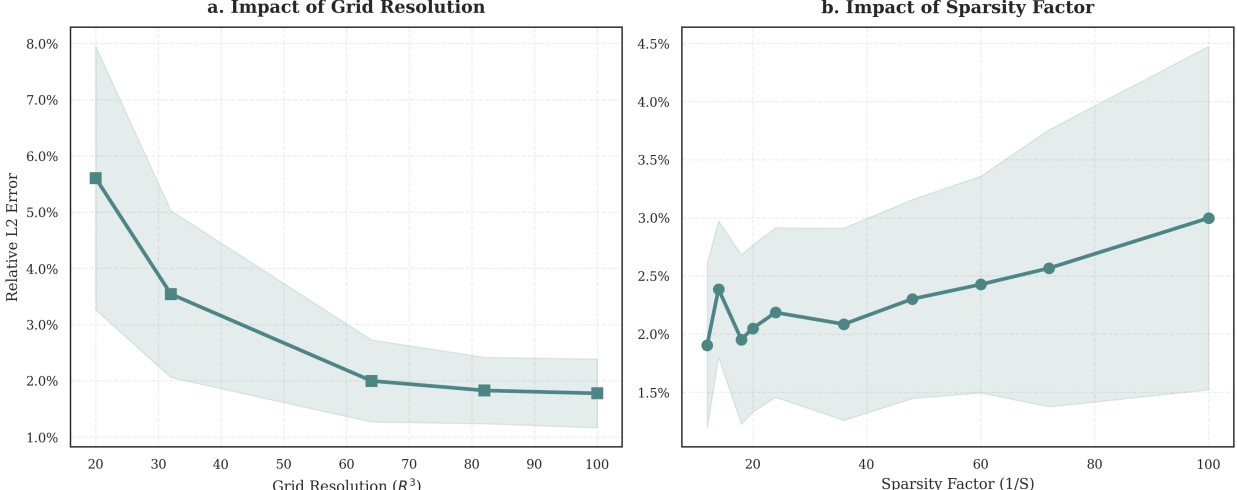

Figure 14: **Ablation study on spatial resolution and sampling density for Elastic dynamics. (a)** Error convergence with respect to grid resolution ($R^3$). The method exhibits rapid convergence, plateauing at $R = 64$, indicating that the $64^3$ grid sufficiently captures the latent deformation manifold. **(b)** Robustness to Lagrangian sparsity at fixed grid size of 64. The model maintains low reconstruction error ($< 3\%$) even when the input particle density is reduced by a factor of 100 (Sparsity Factor $S = 100$), showing that in regimes where the topology is stable, the model is more robust to sparsification. Shaded regions denote $\pm 1$ standard deviation across the test set.

generalize to different sampling strategies but not to different graph structures defined over the same sample size.

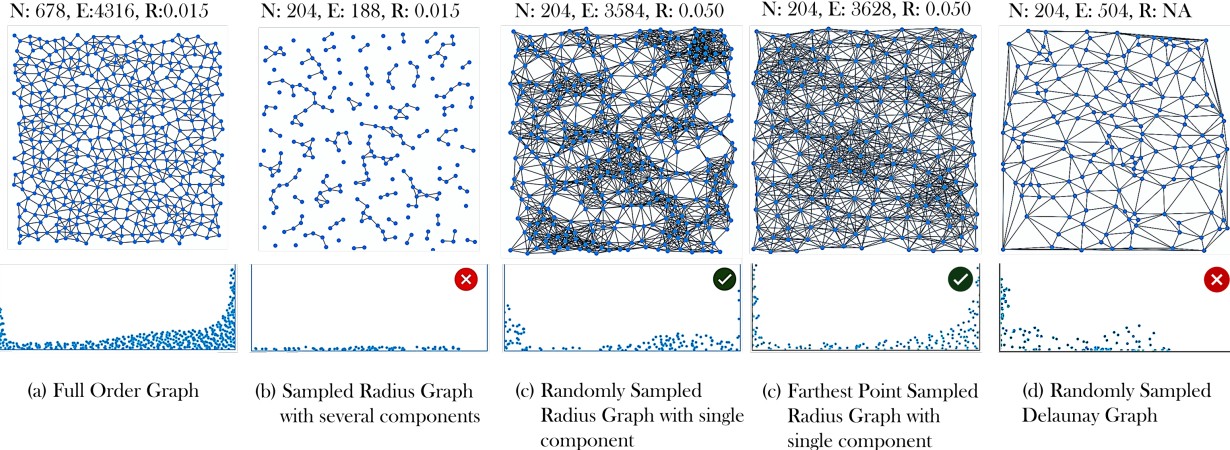

| N: 678, E:4316, R:0.015 | N: 204, E: 188, R: 0.015 | N: 204, E: 3584, R: 0.050 | N: 204, E: 3628, R: 0.050 | N: 204, E: 504, R: NA |

(a) Full Order Graph    (b) Sampled Radius Graph with several components    (c) Randomly Sampled Radius Graph with single component    (c) Farthest Point Sampled Radius Graph with single component    (d) Randomly Sampled Delaunay Graph

Figure 15: This figure illustrates the critical role of radius-based graphs in maintaining discretization invariance and preserving connectivity in graph structures. Both random sampling and farthest-point sampling produce consistent embeddings due to stable neighborhood definitions, whereas altering the graph construction method breaks the model.

### M.3.1 GNNs as Monte-Carlo estimators

We discuss the following propositions made in Li et al. (2020b). Let $D \subset \mathbb{R}^d$ be a compact domain. For each $x \in D$, let $\nu_x$ be a fixed Borel measure on $D$. We define a kernel integral operator of the form:

$$(\mathcal{K}_\phi v)(x) := \sigma \left( W v(x) + \int_D \kappa_\phi(x, y) v(y) \, d\nu_x(y) \right),$$

where $\sigma : \mathbb{R} \to \mathbb{R}$ is a fixed nonlinearity applied elementwise, $W \in \mathbb{R}^{n \times n}$ is a learned matrix, and $\kappa_\phi : \mathbb{R}^{2(d+1)} \to \mathbb{R}^{n \times n}$ is a neural network parameterized by $\phi$. The arguments to $\kappa_\phi$ can include geometric and positional information such as $(x, y, x - y, \|x - y\|)$.

In practice, we do not have access to the continuum $D$, and instead work with a finite point cloud $\{x_1, \ldots, x_K\} \subset D$. We approximate the measure $\nu_x$ by an empirical distribution over a neighborhood $\mathcal{N}(x)$ around each $x$, typically defined via a radius graph. Then, the integral operator is approximated by a sum:

$$(\mathcal{K}_\phi v)(x) \approx \sigma \left( W v(x) + \frac{1}{|\mathcal{N}(x)|} \sum_{y \in \mathcal{N}(x)} \kappa_\phi(e(x, y)) v(y) \right),$$

where $e(x, y)$ denotes edge features constructed from $(x, y)$. This yields the following message-passing update rule:

$$v^{t+1}(x) = \sigma \left( W v^t(x) + \frac{1}{|\mathcal{N}(x)|} \sum_{y \in \mathcal{N}(x)} \kappa_\phi(e(x, y)) v^t(y) \right).$$

We interpret this as a discretization of the continuum operator $\mathcal{K}_\phi$, where the integral is approximated by local aggregation, and the kernel $\kappa_\phi$ is represented by a shared neural network acting on edge features.

The key modeling decision is that $\kappa_\phi$ defines a $K \times K$ block kernel matrix, where each entry $\kappa_\phi(x_i, x_j)$ is a matrix in $\mathbb{R}^{n \times n}$. The parameters $\phi$ are shared across all edge pairs. This sharing ensures that the operator is independent of the number and arrangement of discrete points, so long as the neighborhoods are constructed consistently. Therefore, the learned operator exhibits discretization invariance.

### M.3.2 Discretization or resolution?

Figure 16 presents four point clouds of identical size and geometry but with differing point configurations. Despite many non-overlapping locations, the predictions across all four discretizations remain consistent. This visual evidence supports the notion of learned model generalizing across different samplings of the same underlying domain. It is important to distinguish this from resolution, which concerns the density or number of points used to represent the domain. Discretization refers to the specific arrangement of a fixed number of points; resolution, by contrast, changes the number of points altogether. This is a subtle difference.

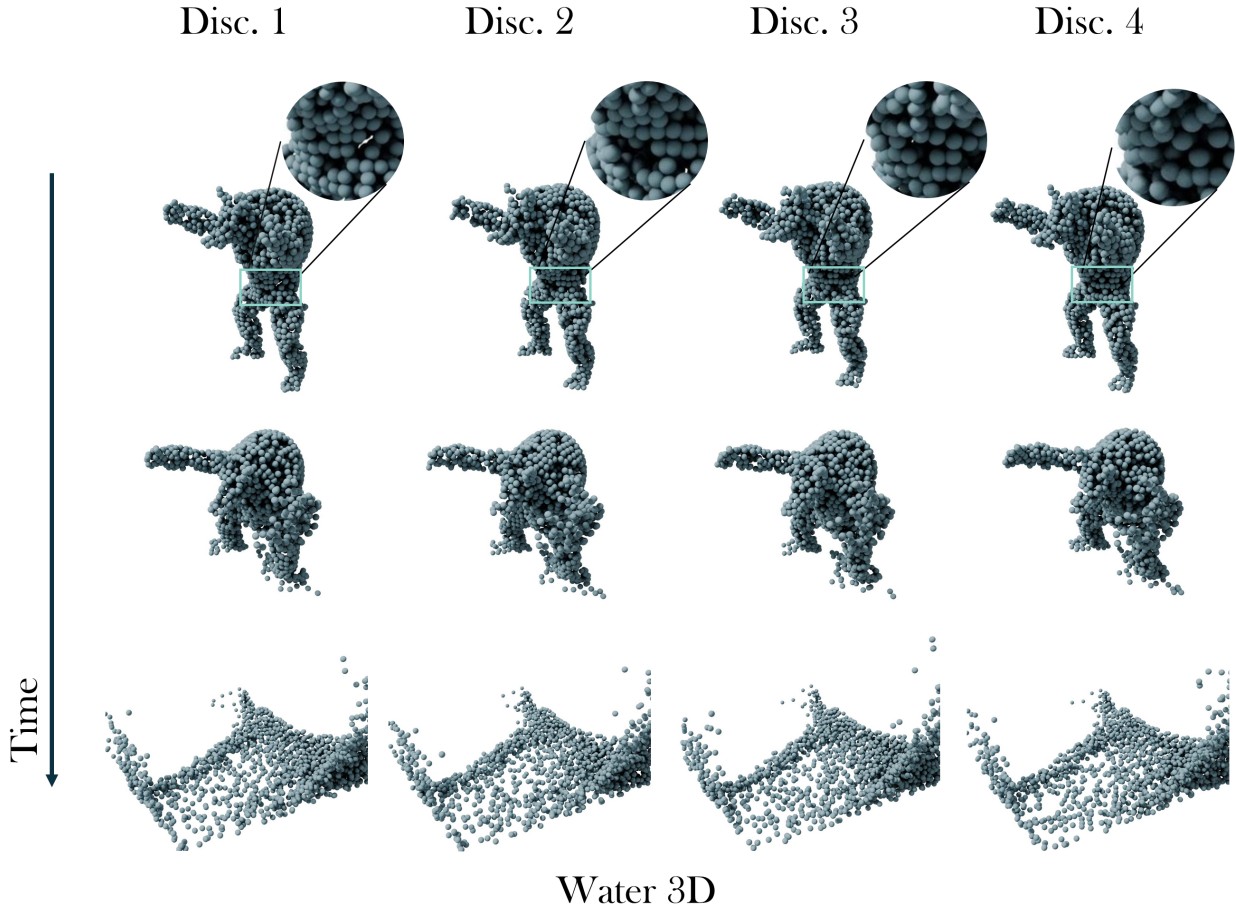

Figure 16: **Discretization invariance on the same geometry.** We construct four different point clouds sampled from the same domain, with equal size but varying locations. On each, we build a radius graph with fixed connectivity radius and apply the same learned GNN layer. Despite the different discrete realizations, the resulting outputs are consistent.

**Implicit handling of self-contact**  Training data were generated using MPM solvers that handle self-contact implicitly via a background Eulerian grid, applicable to both solids and fluids. As a result, the model implicitly learns self-contact behavior from data. Future work could explore improved sampling strategies in regions prone to self-collision for enhanced resolution.

## N   Results

In this section, we present the visual comparisons against baseline ROM architectures.

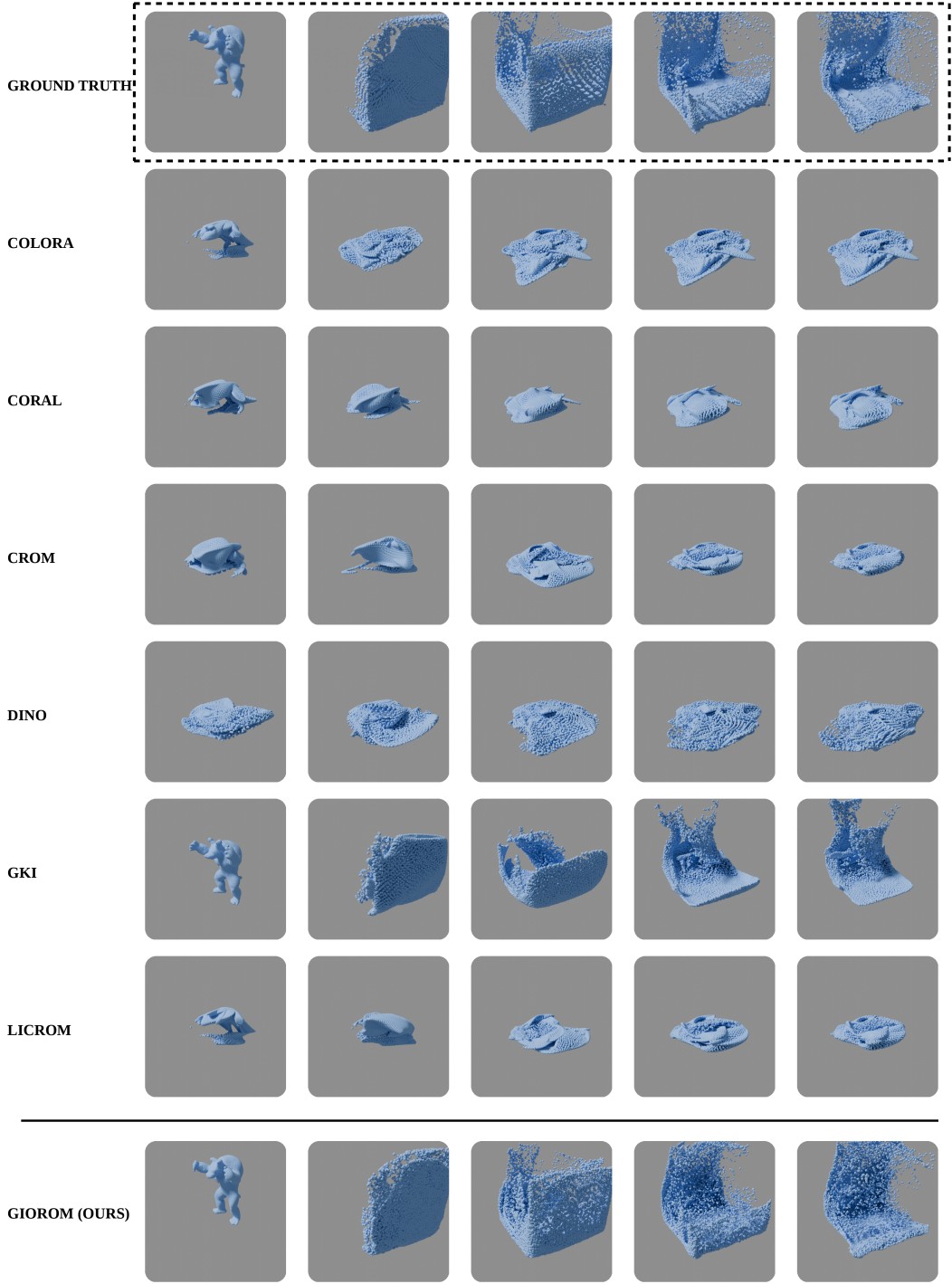

Figure 17: **Comparisons on Water Dataset.** Qualitative evaluation of the reconstructed flow dynamics for the `WATER-3D` dataset. GIOROM effectively preserves fine-scale splashing features compared to global ROM baselines.

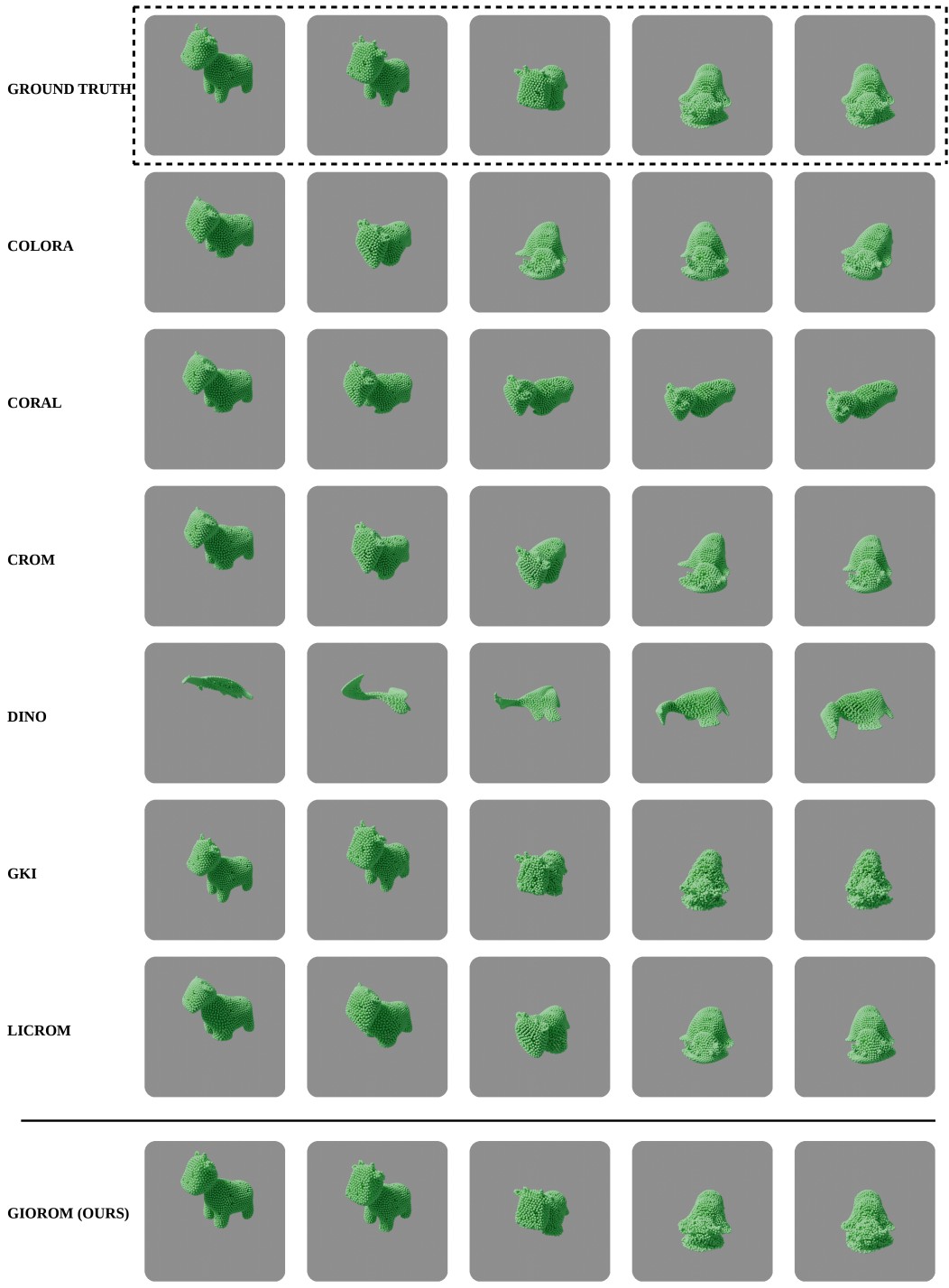

Figure 18: **Comparisons on Plasticine Dataset.** Qualitative results for the `PLASTICINE-3D` dataset, demonstrating the model's ability to handle viscoplastic deformations. GIOROM maintains structural coherence.

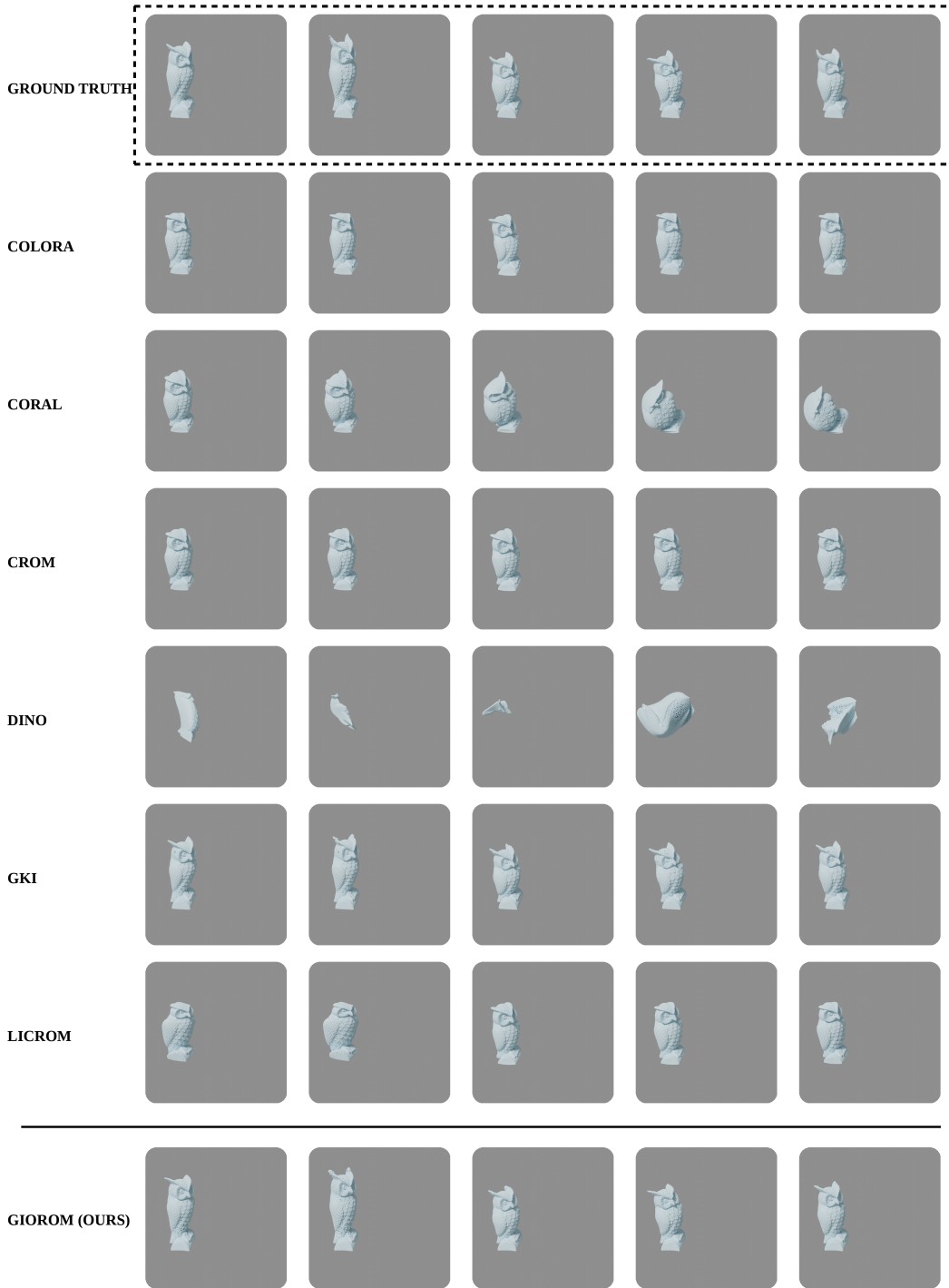

Figure 19: **Comparisons on Elasticity Dataset.** Visual comparison for the `ELASTICITY-3D` (Owl) dataset. The results highlight the method's performance on continuous elastic deformation.

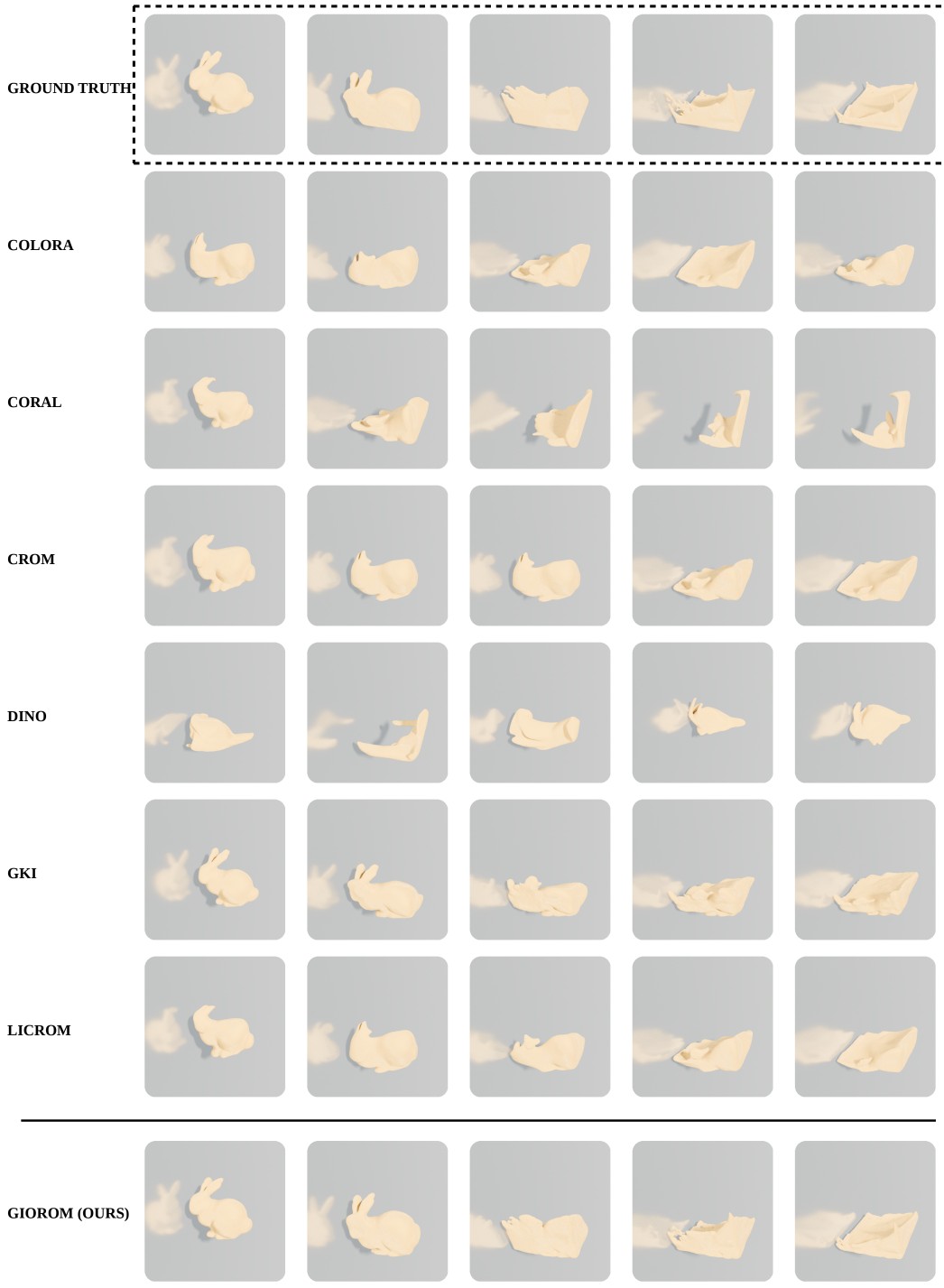

Figure 20: **Comparisons on Granular Sand Dataset.** Evaluation on the `SAND-3D` dataset, characterized by scattering and discrete particle behavior. GIOROM successfully reconstructs the granular flow patterns.

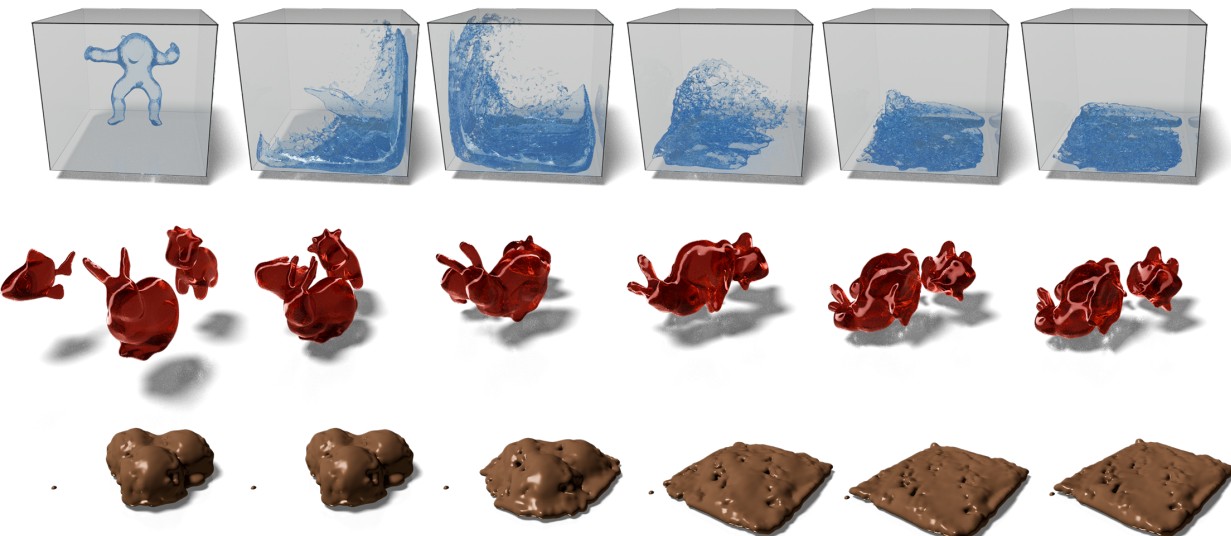

Figure 21: Rollout trajectories for three physical systems evaluated until equilibrium. (**Top**) Water simulation with high dynamism, rolled out for 1000 time steps over 55K particles; rollout MSE: $1.1 \times 10^{-2}$. (**Middle**) Elastic collisions (Jelly) over 500 time steps with 84K particles; rollout MSE: $6.4 \times 10^{-5}$. (**Bottom**) Phase transition in plasticine-like material (Chocolate) over 2000 time steps with 24K particles; rollout MSE: $3.2 \times 10^{-4}$.

Time Stepper
Prediction ➡

Kernel-ROM
Inference ➡

Time Stepper
Prediction ➡

Kernel-ROM
Inference ➡

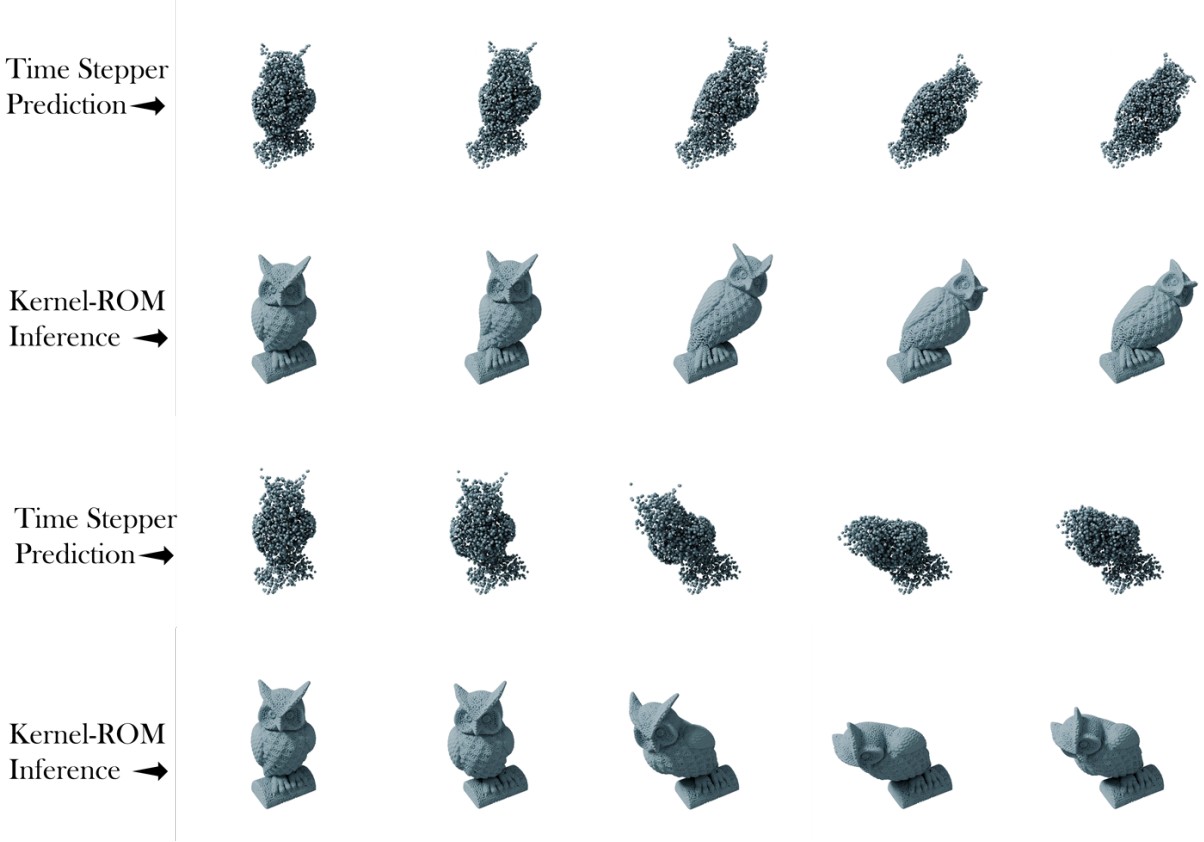

Figure 22: **Subspace vs full-space inference:** Rows 1 and 3 show predictions from the time-stepper on the sampled space; Rows 2 and 4 depict Kernel-ROM output projected back to the full discretization.

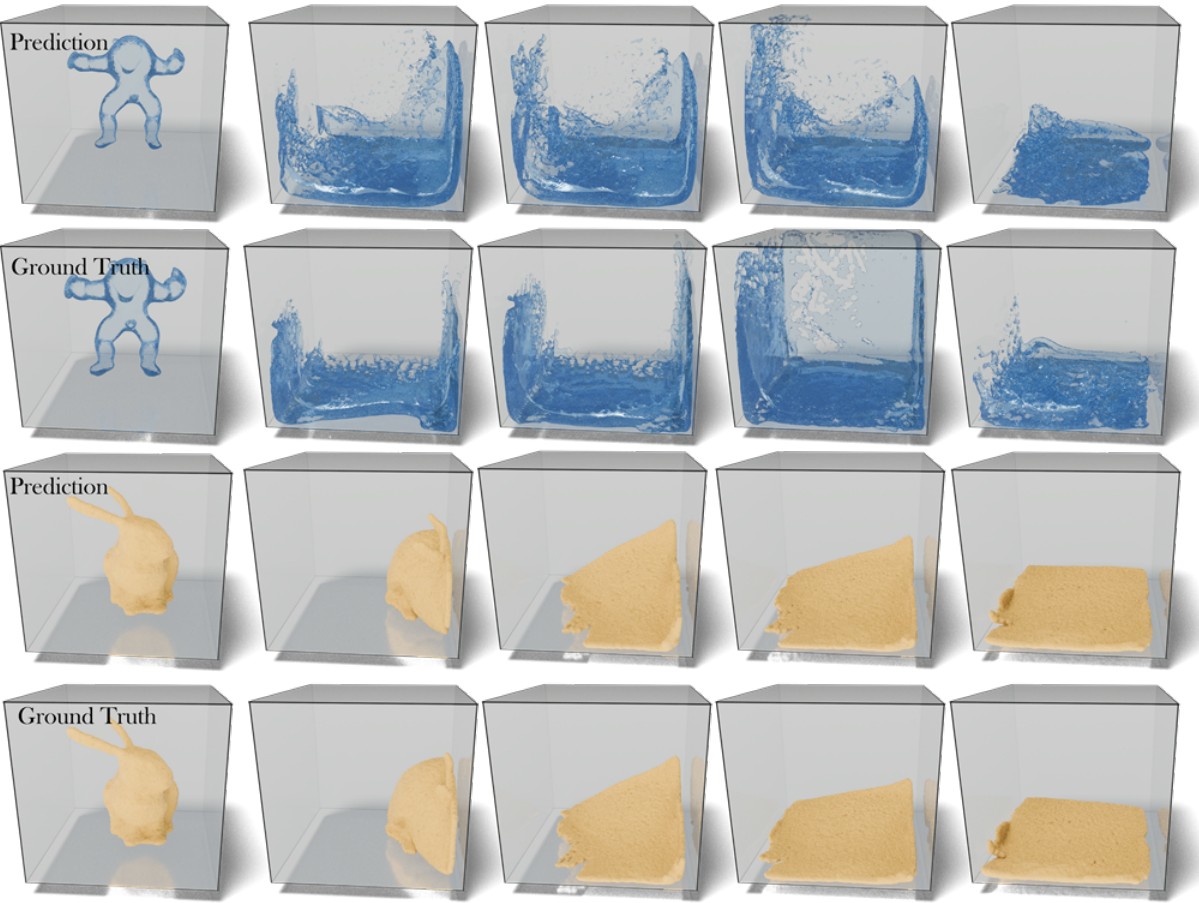

Figure 23: **Ground-truth comparisons:** Rendered outputs for water and sand simulations contrasted against reference solutions.

