# OpenReview forum: "Learning Lagrangian Interaction Dynamics with Sampling-Based Model Order Reduction"
_TMLR — Accepted by TMLR_

### Review · Reviewer_QeMW · 2025-12-16

**Summary Of Contributions:**

The paper introduces GIOROM, a novel sampling-based, discretization-invariant reduced-order modeling framework for Lagrangian physical systems that overcomes key limitations of traditional ROMs and neural PDE solvers. Instead of evolving a global latent state, GIOROM directly evolves a small set of representative particles using a data-driven autoregressive neural time-stepper built from interaction networks and neural operator transformers, drastically reducing active degrees of freedom while preserving local dynamics. To recover full-resolution fields, the method introduces a learnable, locality-aware kernel parameterization that reconstructs the solution manifold at arbitrary spatial points, outperforming projection-based ROMs in highly dynamic regimes such as fluids.

**Additional Comments:**

- In Figure 2, the fluid predictions produced by other baseline methods appear to extend outside the simulation domain. Is this physically plausible, or could it be an artifact of the visualization or model behavior?

- In Figure 3, there appears to be an inverse trend when the sampling ratio decreases from 0.5 to 0.25 in both the “Sparsity vs. GPU” and “Sparsity vs. Time” plots. Could the authors explain the reason for this unexpected behavior?

**Audience:**

Yes

**Audience Explanation:**

Many individuals in TMLR’s audience would likely be interested in the proposed method and its application to the physical systems.

**Broader Impact Concerns:**

There is no ethical concern.

**Claims And Evidence:**

No

**Claims Explanation:**

Table 2 presents the performance of GIOROM on several physical systems. However, the results reported in this table are shown only for GIOROM, without comparison to any baseline methods. As a result, it is unclear whether the reported MSE values are sufficiently good to support the author’s claim that the proposed method is valid and effective for these physical systems. Although the paper discusses a wide range of related work, the experimental comparison is limited. In Table 3, the authors compare their method only against other neural PDE operators, and the evaluation metric is restricted to rollout MSE. This comparison would be more convincing if Table 4 also included computational cost metrics (e.g., runtime or memory usage) alongside MSE. In addition, more qualitative comparisons should be provided through visualizations of predicted trajectories or fields for different methods, rather than limiting such visualization only in Figure 2 (a).

**Requested Changes:**

- In Figure 1, it is difficult to clearly understand the differences between prior ROM techniques and GIOROM. In particular, the distinction between the “PDE stepper” and the “Neural stepper” blocks is unclear, as is the role and input of the decoder g and the kernel network k. The purpose of the dotted-line box on the right-hand side of the figure should also be more explicitly explained.

- Additional ROM baselines, such as CROM, should be included in Table 2 for a more comprehensive comparison. Moreover, the effect of sparsity on rollout error, as shown in Figure 3, should also be compared against these alternative ROM methods.

---

> ### Author Response · Authors · 2026-01-18
>
> We appreciate the reviewer for providing constructive feedback on our work and we provide the answers for the queries.
>
> **The results reported in this table are shown only for GIOROM, without comparison to any baseline methods.**
>
> We have fixed this issue. For the end-to-end generation, we provide a comparison against SOTA full-order GNN based lagrangian solver (GNS) [1]. This table computes the end-to-end performance (i.e. sample space time-stepping + Full space inference). We have moved the ablation on sample space time-stepping (i.e. comparing just the time-stepper with baselines GNOT, GNO, etc. to the appendix for space constraints). Table 2 is now reordered  to Table 3.
>
> **the experimental comparison is limited. In Table 3, the authors compare their method only against other neural PDE operators, and the evaluation metric is restricted to rollout MSE. This comparison would be more convincing if Table 4 also included computational cost metrics (e.g., runtime or memory usage) alongside MSE., lack of qualitative results**
>
> We completely agree with the reviewer on all of the above points. As such we have done our best to do a fully comprehensive quantitative as well as qualitative analysis against SOTA ROM methods. Table 4 has been moved to appendix and serves as the ablation on neural PDE operators. We instead present a comprehensive ROM baseline covering a range of Lagrangian ROM baselines - neural ODE based DINo, CORAL, Projection based CROM, LiCROM, Graph neural operator based GKI, Low rank model CoLORA, and classical PCA. We consider several metrics - Relative Error, along with the standard deviation, average Chamfer Distance, average memory consumed per simulation, peak memory consumed per simulation and total wall clock inference time. Table 2 provides quantitative numbers for these metrics. Furthermore, we provide Qualitative metrics for 4 diverse regimes for all the above models in Figures 16, 17, 18, 19. We believe these to be comprehensive, however, we are happy to provide additional metrics.
>
> **In Figure 1, it is difficult to clearly understand the differences between prior ROM techniques and GIOROM. In particular, the distinction between the “PDE stepper”**
>
> We have added clarifications in the caption to address these issues. The diagram follows the architecture setups used in CROM[2], MPM[3]. We provide additional implementation details in Section 3.
>
> **the effect of sparsity on rollout error, as shown in Figure 3, should also be compared against these alternative ROM methods.**
> We provide these in Figure 5, where we contrast the reduced space size (sparsity) against other ROM architectures and provide a discussion on failure modes of all the models.
>
> **In Figure 2, the fluid predictions produced by other baseline methods appear to extend outside the simulation domain. Is this physically plausible, or could it be an artifact of the visualization or model behavior?**
>
> As these are fully data-driven models, the entire spatial field is compressed into a single global latent vector and as such, the local spatial information, including boundary conditions are not captured in this regime, resulting in boundary violations.
>
> **In Figure 3, there appears to be an inverse trend when the sampling ratio decreases from 0.5 to 0.25 in both the “Sparsity vs. GPU” and “Sparsity vs. Time” plots. Could the authors explain the reason for this unexpected behavior?**
>
> The inverse trend is observed due to the increase in radius as the sampling ratio decreases. To avoid neighborhood collapse, we tune the radius to ensure full coverage within the graph. This leads to a local increase in sample count and a subsequent increase in GPU.
>
> [1] Sanchez-Gonzalez, Alvaro, et al. "Learning to simulate complex physics with graph networks." International conference on machine learning. PMLR, 2020.
>
> [2] Chen, Peter Yichen, et al. "CROM: Continuous reduced-order modeling of PDEs using implicit neural representations." arXiv preprint arXiv:2206.02607 (2022).
>
> [3] Chen, Peter Yichen, et al. "Model reduction for the material point method via an implicit neural representation of the deformation map." Journal of Computational Physics 478 (2023): 111908.

---

### Review · Reviewer_ZH43 · 2025-12-31

**Summary Of Contributions:**

This paper proposes a method called GIOROM (Geometry-informed reduced order modelling), which is a framework for simulating Lagrangian particle dynamics in physical systems. The key idea is to replace the traditional projection-based reduced-order modelling with a sampling-based approach that evolves dynamics in physical space with a learnable kernel parameterization.

Strengths:

1. The method introduced seems novel in that it combines reduced-order modeling with direct physical space evolution, avoiding global latent representations that struggle with localized dynamics.

2. Empirical results reported are convincing.

3. Experiments are reported on multiple material types with detailed ablations studying sparsity effects, sampling strategies, and architectural choices.

4. The approach is agnostic to underlying neural operator architecture, making it broadly applicable.

Weaknesses:

1. Paper lacks theoretical justification about the following: Convergence guarantees for the kernel approximation, Error bounds as a function of sampling density and/or justification for when the method should work.

2. Extreme sparsity failure is poorly characterized.

3. The reported baselines are rather limited ROM baselines are under-tested, and no comparison with recent learning-based ROM methods (beyond 2023).

4. Computational cost analysis is incomplete in that the memory scaling with radius could be prohibitive and there is no wall-clock time comparisons for complete simulations.

5.  Experiments on only 100 trajectories per system seem modest for deep learning

6. There is no discussion of generalization to different boundary conditions or domain sizes.

7. Writing can be made a little lighter.

**Audience:**

Yes

**Audience Explanation:**

This paper deals with an important sub-area of deep learning research, namely, the neural PDE operators, and therefore is of interest to the community.

**Claims And Evidence:**

Yes

**Claims Explanation:**

The claims made are failry supported with empiral evidence (some gaps exists, see questions).

**Requested Changes:**

1. Provide empirical lower bounds on sampling density as a function of system properties (e.g., velocity gradients, material stiffness, geometry complexity).

2. Please consider testing all ROM methods across all material types, not just water, by including recent neural ROM methods.

3. Consider adding a table showing when GIOROM vs. other methods is preferable.

4. Even approximate error bounds or convergence conditions would strengthen claims significantly.

5. Presentation can be reorganized -  critical figures (8-9) to can be moved to the main paper

6. Code and datasets should be released for reproducibility along with hyperparameter selection strategies.

7. Consider discussions/ demonstration of adaptive time-stepping.

8. Fix notation inconsistencies.

9. Discuss computational requirements for graph construction overhead.

10. Provide ablation on the window size for velocity sequences.

---

> ### Author Response · Authors · 2026-01-18
> **Author response**
>
> We appreciate the reviewer for providing constructive feedback in our work. We have addressed all of the above concerns in the updated pdf. All the changes are highlighted in red in the updated manuscript. Here are the answers to the questions
>
> **Extreme sparsity failure is poorly characterized and The reported baselines are rather limited ROM baselines are under-tested, and no comparison with recent learning-based ROM methods (beyond 2023).**
>
> We agree with the reviewer on these comments and apologize for their exclusion. We have attempted to cover all the recent ROM architectures with openly available codebase. As such in the updated manuscripts (Table 2, Figure 5, Figures 16-19), we compare both qualitatively and quantitatively against the following models  - NODE based DINO, CORAL, Low rank based CoLORA, projection based CROM, LICROM, Classical PCA, graph neural operator based GKI.
>
> We additionally discuss stability over sparsification in Appendix E, L and M.
>
> **Computational cost analysis is incomplete in that the memory scaling with radius could be prohibitive and there is no wall-clock time comparisons for complete simulations.**
>
> In the updated manuscript, we provide new experiments to study the impact of scaling with the size of the reduced space. We specifically compare the impact of constructing large graphs for both Kernel-integral ROM as well as time-stepper. We point to figure 3 where we provide separate wall clock times for graph construction as well as kernel regression. We show scaling in memory as well as compute time. We provide graphs showcasing both radius as well as sample size. For the time-stepper, as we use several Interaction layers, we are unable to compute individual graph processing time and as such in Figure 4, we show the end-to-end wall clock time for a single forward pass. Additionally in appendix L and M, we perform empirical analysis of sparsification on the performance of graph based time-stepper, considering two sampling regimes at fixed radius, on two materials (Water and Plasticine). We also compute the impact of radius at different sampling regimes (Figures 10, 11). We provide additional implementation details in Section 3.
>
> **There is no discussion of generalization to different boundary conditions or domain sizes.**
>  As our approach is end-to-end data-driven, we do not encode any PDE based constraints. However, in Appendix G, we discuss implicit boundary encoding, which we implement following GNS[1]. This provides a data-driven approximation of the boundary condition, allowing the model to learn directly from the data. As such, we show that the model generalizes to different domain sizes, via. Sampling (Figure 2b), but we do not explicitly enforce boundary conditions.
>
> **Requested changes 1-4**
> We have done our best to provide all of the requested changes. Figure 5 provides empirical lower bounds on ROM reduced space against performance over 4 different materials. We have provided a table with quantitative results (Table 2) and qualitative results in Figs 16-19. We are happy to provide additional empirical results/clarifications.
>
> **Code and datasets should be released for reproducibility along with hyperparameter selection strategies.**
> We have provided a demo executable code and a small sample dataset (2 shortened trajectories, over a small domain) due to space restrictions. With this code, we demonstrate the performance of the kernel-integral ROM. However, we do not currently provide the code for the time-stepper as our approach is invariant to the underlying model and any SOTA model such as GNS [1] may be leveraged. We have updated the appendix to reflect ablations on hyperparameters.
>
> **Consider discussions/ demonstration of adaptive time-stepping**
> In our current dataset and experimental setting, we are restricted to consistent time discretization, limited by the architectures of neural time steppers. However, it is an area of future work, which we discuss in the last section of the paper. We have provided this discussion in the concluding sections of the paper
>
> **Provide an ablation on the window size**
> In this work, we focus on consistent time parameterization and leverage the interaction network proposed in GNS[1], and as such we follow the same hyperparameters for the graph based layers as in GNS and as such we do not observe any differences compared to GNS, as the encoder layers are identical.
>
> [1] Sanchez-Gonzalez, Alvaro, et al. "Learning to simulate complex physics with graph networks." International conference on machine learning. PMLR, 2020.

---

### Review · Reviewer_nUUM · 2026-01-08

**Summary Of Contributions:**

This paper proposes a neural approximation framework for simulating physical systems governed by Lagrangian interaction dynamics, aiming to improve efficiency at high spatial resolutions. While many existing neural solvers adopt a reduced-order modeling (ROM) paradigm that evolves a low-dimensional latent representation, such global projection-based approaches can struggle to capture localized, rapidly evolving dynamics. The authors instead evolve a sparsely subsampled set of particles directly in physical space using a learned neural time-stepper, and reconstruct the full field via a learnable, locally defined kernel-based estimator. By decoupling reduced-order temporal evolution from spatial reconstruction, the method preserves locality while substantially reducing the number of active degrees of freedom during simulation. Experiments across several Lagrangian systems, including fluids and elastoplastic materials, demonstrate favorable accuracy–efficiency trade-offs and stable long-horizon rollouts.

**Audience:**

Yes

**Audience Explanation:**

Neural operators for approximating PDE solvers and data-driven PDE modeling constitute a rapidly growing area of research, with broad potential applications across many domains, particularly in AI for Science. Addressing scalability to high-resolution fields remains one of the most important practical bottlenecks in this area, and the motivation of the paper directly targets this challenge.

**Broader Impact Concerns:**

I don't find any broader impact concern for this paper.

**Claims And Evidence:**

Yes

**Claims Explanation:**

The experiments show improved accuracy over ROM baselines as well as clear computational efficiency gains, providing convincing support for the paper’s claims.

**Requested Changes:**

- The paper motivates the proposed approach by arguing that reduced-order modeling (ROM) methods are inherently prone to losing important local information due to projection onto low-dimensional spaces. It would be helpful for the authors to elaborate on this claim. In particular, is this criticism intended to apply only to the specific ROM baselines considered in the paper (e.g., PCA- or autoencoder-based global latent models), or does it extend more broadly to latent space learning approaches in general? In many other areas of machine learning, learning continuous latent representations has proven highly effective, with sufficient flexibility to reconstruct fine-grained local details. Clarifying whether and why such representations are especially challenging to employ for physical systems governed by Lagrangian dynamics would strengthen the motivation and positioning of the proposed method.

- In Figure 3 (upper-left and upper-right), both GPU usage and runtime exhibit noticeable non-monotonic behavior, with apparent outliers around the 0.25× sampling ratio. Could you clarify the source of these bumps?

- Is the Hilbert space assumption valid for practical scenarios?

---

> ### Author Response · Authors · 2026-01-18
>
> We appreciate the reviewer for providing constructive feedback. We provide the answers to the questions below
>
> **learning continuous latent representations has proven highly effective, with sufficient flexibility to reconstruct fine-grained local details. Clarifying whether and why such representations are especially challenging to employ for physical systems governed by Lagrangian dynamics would strengthen the motivation and positioning of the proposed method**
>
> In the updated manuscript, we have done our best to provide comparisons against publicly available SOTA latent learning methods. We have updated Table 2, Figs 16-19 to show both quantitatively and qualitatively how several of these models struggle with highly dynamic lagrangian systems. In particular we have considered NODE based DiNO, CORAL, Low rank based CoLORA, projection based CROM, LiCROM, graph neural operator based GKI, and PCA.
>
> **In Figure 3 (upper-left and upper-right), both GPU usage and runtime exhibit noticeable non-monotonic behavior, with apparent outliers around the 0.25× sampling ratio. Could you clarify the source of these bumps?**
>
> the radius is tuned for extremely dense systems to avoid crashes due to OOM issues. As such, in sparser regimes, we increase the radius to avoid collapse of the graph topology (as seen in Figure 13), this optimization allows the system to function, but causes a minor bump. However, in the subsequent regions, the graph remains stable. As we show in bottom right, increasing radius arbitrarily does not however affect the performance.
>
> **Is Hilbert space assumption valid in practical scenarios**
> Yes, for the cases considered in this work, this mathematical assumption just requires that the total "energy" of the system is finite. Since our experiments run inside a fixed box (a bounded domain) and physical values like speed and density never become infinite, the data naturally fits the requirements. However, modeling explicit discontinuities or free flow scenarios is a consideration for future work.

---

### Decision · Action_Editor_LVVJ · 2026-02-24

**Recommendation:** Accept as is

**Audience:**

Yes

**Audience Explanation:**

Yes, the paper addresses a practical bottleneck (scalability of neural PDE solvers to high-resolution domains) that is broadly relevant to the AI-for-science and scientific computing communities well represented in TMLR's readership.

**Claims And Evidence:**

Yes

**Claims Explanation:**

Thank you to the reviewers for their thorough evaluation and to the authors for their comprehensive revisions. All three reviewers converged on Leaning Accept after the rebuttal, acknowledging that the revised manuscript addresses the main concerns raised during review; notably the addition of extensive ROM baselines, wall-clock and memory comparisons, and improved clarity of presentation. The paper presents a well-motivated and empirically validated framework for sampling-based reduced-order modeling of Lagrangian systems, with convincing results across multiple material types. While the contribution is primarily empirical and the technical novelty is somewhat incremental, the work is correct, clearly relevant to the community, and meets the acceptance criteria for TMLR. I recommend acceptance without certifications.